# Normalization in Attention Dynamics

Nikita Karagodin[1]    Shu Ge[2]    Yury Polyanskiy[1]    Philippe Rigollet[2‡]

## Abstract

We study the effect of normalization schemes on token representations in deep transformers. Modeling their evolution as interacting particles on the sphere, we show that normalization acts as a form of speed regulation. This perspective enables a unified analysis of several schemes—including **Post-LN**, **Pre-LN**, **Mix-LN**, **Peri-LN**, **nGPT**—revealing how they influence clustering dynamics and representation collapse. Our framework clarifies how different schemes shape token representations across layers and provides a principled basis for comparing them, identifying **Peri-LN** as a particularly effective choice.

## 1  Introduction

Transformer architectures have revolutionized natural language processing and beyond, demonstrating unprecedented performance across diverse tasks—from machine translation and text generation to reasoning and protein folding. The remarkable capabilities of transformers, including their emerging reasoning abilities, are enabled by the attention mechanism introduced in Bahdanau et al. (2015); Vaswani et al. (2017).

A recent line of theoretical work, initiated in Geshkovski et al. (2023), studies information processing across deep transformer layers by reframing them as interacting particle systems, building on the original setup of Sander et al. (2022). Following this initial work, layer normalization (LayerNorm) emerged as a critical component significantly influencing the long-term dynamics of these systems. Geshkovski et al. (2025) proposed a model in which particles are constrained to evolve on a sphere, corresponding to the so-called *Post-layer norm* (**Post-LN**) scheme. This model has since become the standard paradigm for transformer analysis in subsequent research (Karagodin et al., 2024; Geshkovski et al., 2024a,b; Bruno et al., 2025a,b; Criscitiello et al., 2024).

Several alternatives to Post-LayerNorm (**Post-LN**) have emerged in recent years to improve training performance, each subtly altering transformers' long-term clustering behavior. Most notably, Pre-LayerNorm (**Pre-LN**) has become the default choice for leading large language models including GPT (Radford et al., 2019) and LLaMA (Touvron, H. et al, 2023). This approach was originally introduced in ResNet-v2 He et al. (2016) before being adapted for Transformer architectures. It enables more stable training of deeper networks while reducing sensitivity to hyperparameters such as learning rates (Xiong et al., 2020).

Understanding normalization schemes is essential for advancing the design and performance of transformer architectures. In particular, Sun et al. (2025) and Gromov et al. (2025) identify a phenomenon known as the *curse of depth*, in which deep layers of large language models (LLMs) degenerate into near-identity transformations. This effect is so pronounced that it enables pruning of deeper layers with minimal impact on performance (Muralidharan et al., 2024; Siddiqui et al., 2024). On the other hand, the well-known issue of *representation collapse* presents a significant challenge to increasing the depth of LLMs.

---

[*1]Department of EECS, MIT, Cambridge, MA, USA
[†2]Department of Mathematics, MIT, Cambridge, MA, USA

39th Conference on Neural Information Processing Systems (NeurIPS 2025).

To mitigate this issue, Li et al. (2025) propose a hybrid normalization scheme that applies **Post-LN** normalization in the early layers and reserves **Pre-LN** normalization for the deeper layers. This strategy was further refined in the development of **Peri-LN** (Kim, B., Johnson, M. et al., 2025), which has been reported to be used in the Gemma-3 model (Gemma Team et al., 2025). Alternatively, Noci et al. (2022) suggest a simpler fix: rescaling residuals by the square root of the depth (a method we will call **sqrt-scaling**). This method with **Pre-LN** architecture was then studied in-depth by Sun et al. (2025). Additionally, Loshchilov et al. (2025) show that with careful architectural design, as in **nGPT**, normalizing tokens to lie on the unit sphere can further streamline the normalization process.

Given the diversity of these approaches, we are motivated to explore the following question:

*How do normalization schemes influence deep representations in transformers?*

To answer this question, we revisit both classical and novel LayerNorm schemes through the lens of the simplified interacting particle dynamics introduced in Geshkovski et al. (2025) to bring a theoretical understanding of these various design choices. Since the final decoding layer of a transformer is typically preceded by a normalization step, we focus on the *direction* of token representations. Regardless of the specific normalization used, these directions naturally form an interacting particle system on the sphere. This shared geometric setting enables a direct, side-by-side comparison of various normalization schemes, all of which we reinterpret as forms of *speed regulation*. Despite its simplicity, our model captures complex behaviors observed in practice, including *curse of depth* and *representation collapse*.

**Related Work.** A growing body of work has examined normalization in Transformers, with a primary focus on its empirical and theoretical implications for gradient stability. Notably, Xiong et al. (2020) and Sun et al. (2025) provide experimental evidence that improper placement of normalization layers can lead to exploding or vanishing gradients in deep models. These findings are often supported by variance-based analyses that track the propagation of activations and gradients through the network, such as (Noci et al., 2022) and (Kedia et al., 2024). Wortsman et al. (2024) further identify normalization-related training instabilities that emerge at scale. Building on this foundation, Li et al. (2025) and Kim, B., Johnson, M. et al. (2025) explore hybrid normalization strategies in large-scale settings, using both theoretical approximations and empirical diagnostics to study gradient flow and the stability of learned representations.

In contrast to prior work that primarily investigates gradient dynamics, our study focuses on the forward evolution of token representations through the network. This perspective complements the analysis of gradient flow by shifting the emphasis from the ability to train (via backpropagation) to the expressiveness and structure of the learned representations. While both viewpoints offer valuable insights, we focus on the latter in the present work. A companion paper dedicated to the analysis of gradients is currently in preparation.

**Our contributions.** We provide different perspectives on normalization architecture, by casting differently normalized Transformers as variations of a common interacting-particle ODE, where the normalization method determines a *speed factor*, which can amplify initial velocity and dampen representation collapse in deep layers. Within this unified framework, we extend the framework of Geshkovski et al. (2025) for **Post-LN** and in particular, establish asymptotic clustering under general conditions on the speed regulation mechanism. To differentiate various normalization schemes we further study the initial and final velocity of tokens corresponding to first and deep layers respectively. In particular, we recover the representation collapse phenomenon that plagues **Post-LN**. Our theoretical framework identifies **Peri-LN** as a particularly effective scheme that makes good use of both early and deep layers.

## 2 Normalized Attention Dynamics

A sequence of $n$ tokens is represented by their column-vector embeddings $X = (x_1, \ldots, x_n) \in \mathbb{R}^{d \times n}$. In the rest of this section, functions $f : \mathbb{R}^d \to \mathbb{R}^d$ applied to such a matrix are understood column-wise: $f(X) = [f(x_1), \ldots, f(x_n)]$. For each token embedding $x_k$, we define its *direction* $\theta_k = x_k/\|x_k\| \in \mathbb{S}^{d-1}$ and its *magnitude* $r_k = \|x_k\| \geq 0$, so that

$$x_k = r_k \cdot \theta_k.$$

As the sequence of token embeddings is processed through the layers of a transformer, it gets updated from $X^t$ to $X^{t+1}$ at layer $t$. In the rest of this section we derive the updates obtained by different normalization rules and recast them as speed regulation mechanisms for token directions.

For simplicity and convenience of exposition, we omit FFN layers and focus on pure attention. The approach could be extended to a more general architecture, but this would introduce additional technical complexities beyond the scope of this paper.

## 2.1 Attention

At layer $t$, an attention head is characterized by three matrix parameters $Q^t, K^t, V^t$, called Query, Key, and Value respectively. These matrices are used to create the *attention matrix*, which is an $n \times n$ matrix $W = \{w_{jk}\}_{1 \leq j,k \leq n}$ of pairwise interactions between tokens with entries given by

$$w_{jk}^t = \frac{e^{\beta \langle Q^t x_j, K^t x_k \rangle}}{\sum_{l=1}^n e^{\beta \langle Q^t x_j, K^t x_l \rangle}},$$

where we added a redundant temperature parameter usually taken equal to 1 but that will be convenient in our simplifications below. The attention function is the linear operator $A^t : \mathbb{R}^{d \times n} \to \mathbb{R}^{d \times n}$ defined as $A^t(X) = [X_1^t(X), \dots, A_n^t(X)]$ where each column is given by

$$A_j^t(X) = \sum_{k=1}^n w_{jk}^t V^t x_k, \qquad j = 1, \dots, n.$$

Throughout this paper, we focus on the simplified setting of Geshkovski et al. (2025) where $Q^t = K^t = V^t = I_d$ for all $t \geq 0$.

## 2.2 Normalization.

The Root Mean Squared (RMS) norm $\mathsf{Norm}(x) = x/\|x\|$ of a token Zhang and Titov (2019) is a critical ingredient of all normalization schemes considered here.

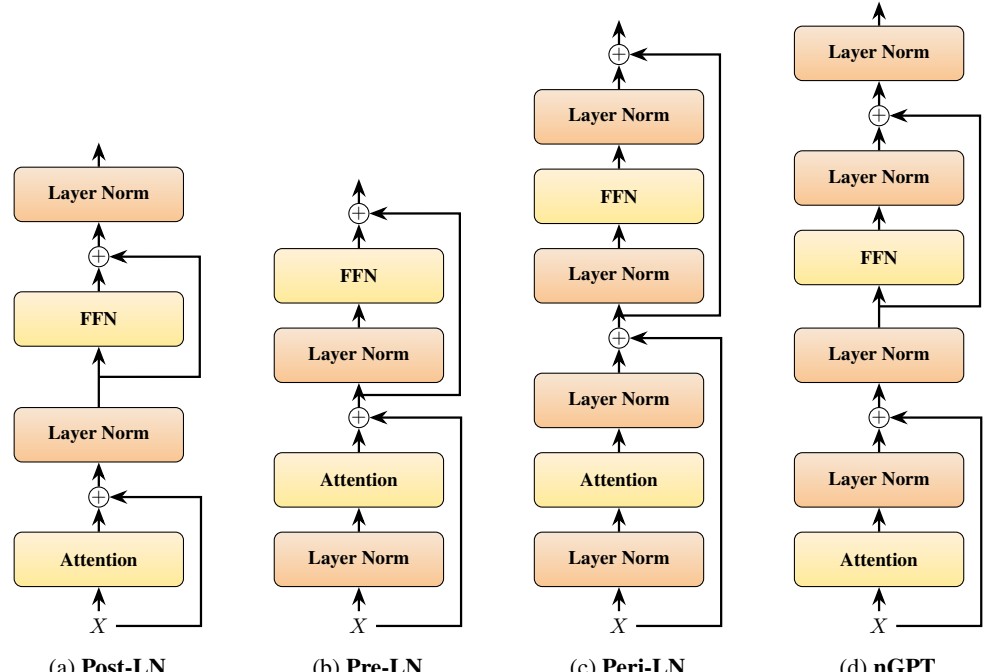

(a) **Post-LN**    (b) **Pre-LN**    (c) **Peri-LN**    (d) **nGPT**

Figure 1: Normalization layer placements in various architectures.

In this paper, we study six major schemes: **Post-LN** (Vaswani et al., 2017), **Pre-LN** (Xiong et al., 2020), **Mix-LN** (Li et al., 2025), **Peri-LN** (Kim, B., Johnson, M. et al., 2025), **nGPT** (Loshchilov

et al., 2025), and **sqrt-scaling** (Sun et al., 2025). Note that **Mix-LN** is a combination of **Post-LN** for $t \leq \tau$ and **Pre-LN** for $t > \tau$ while **sqrt-scaling**is a deterministic rescaling of **Post-LN**. The four remaining schemes are presented in Figure 1. Such explicitly layer-normalization rules are not the only strategies employed in practice. Other attempts to improve normalization suggest better initializations $Q^0, K^0, V^0$ (Kedia et al., 2024) and explicit scaling of the updates, similarly to $\alpha_t$ that is trainable in **nGPT**.

Thanks to the residual connections, each layer-update can be seen as a forward Euler discretization of a continuous-time ODE that captures the dynamics of tokens while enabling the deployment of useful calculus tools. In this context, it is convenient to write $X(t)$ as a function of time and replace $X^{t+1} - X^t$ with $\dot{X}(t)$. For any two matrices $X, Y \in \mathbb{R}^{d \times n}$ where $X$ has unit-norm columns, define the projection operator $\mathbf{P}_X Y$ to be the column-wise projection on to the tangent space of the sphere $\mathcal{S}^{d-1}$:

$$\mathbf{P}_X Y = \left[ \mathbf{P}_{x_1} y_1, \ldots, \mathbf{P}_{x_n} y_n \right],$$

where for any $x \in \mathcal{S}^{d-1}, y \in \mathbb{R}^d$, $\mathbf{P}_x y = y - \langle y, x \rangle x$ is the projection of $y$ onto the tangent space of $\mathcal{S}^{d-1}$ at $x$.

The dynamics described by each normalization schemes are presented in Table 1.

Table 1: Normalization Schemes in Discrete and Continuous Time Domains. In **nGPT**, $\alpha_t \in \mathbb{R}$ is a layer-dependent learnable parameter.

| Scheme | Discrete Time Update | Continuous Time Update |
|---|---|---|
| **Post-LN** | $X^{t+1} = \text{Norm}\big(X^t + A^t(X^t)\big)$ | $\dot{X}(t) = \mathbf{P}_{X(t)} A^t(X(t))$ |
| **Pre-LN** | $X^{t+1} = X^t + A^t\big(\text{Norm}(X^t)\big)$ | $\dot{X}(t) = A^t\big(\text{Norm}(X(t))\big)$ |
| **Mix-LN** | $X^{t+1} = \big[\text{Norm}\big(X^t + A^t(X^t)\big)\big] \mathbb{1}_{t \leq \tau}$ $+ \big[X^t + A^t\big(\text{Norm}(X^t)\big)\big] \mathbb{1}_{t > \tau}$ | $\dot{X}(t) = \big[\mathbf{P}_{X(t)} A^t(X(t))\big] \mathbb{1}_{t \leq \tau}$ $+ \big[A^t\big(\text{Norm}(X(t))\big)\big] \mathbb{1}_{t > \tau}$ |
| **Peri-LN** | $X^{t+1} = X^t + \text{Norm}\big(A^t(\text{Norm}(X^t))\big)$ | $\dot{X}(t) = \text{Norm}\big(A^t(\text{Norm}(X(t)))\big)$ |
| **nGPT** | $X^{t+1} = \text{Norm}\big(X^t + \alpha_t \text{Norm}(A^t(X^t))\big)$ | $\dot{X}(t) = \mathbf{P}_{X(t)} \alpha_t \text{Norm}(A^t(X(t)))$ |
| **sqrt-scaling** | $X^{t+1} = \text{Norm}\big(X^t + \frac{1}{\sqrt{t+1}} A^t(X^t)\big)$ | $\dot{X}(t) = \frac{1}{\sqrt{t+1}} \mathbf{P}_{X(t)} A^t(X(t))$ |

### 2.3 Speed regulation formulation

In **Post-LN**, **nGPT**, and **sqrt-scaling**, tokens are constrained to the the sphere $\mathcal{S}^{d-1}$ with **sqrt-scaling** simply adjusting the speed of the particles as a function of $t$ compared to **Post-LN**. For the other rules where tokens may have varying magnitude, one final projection is typically applied before the final decoding layer in practice. In particular, this means that decoding depends on *directions* $\theta_j(t) = \text{Norm}(x_j(t))$, for $j = 1, \ldots, n$.

Interestingly, when tracking only the directional components $\theta_1(t), \ldots, \theta_n(t) \in \mathcal{S}^{d-1}$, all normalization rules give rise to interacting particle systems evolving on the sphere, governed by a common velocity field but subject to distinct, rule-dependent speed-regulation mechanisms. Note that this does not imply the particles follow the same trajectories at different speeds; indeed the speed parameter has a significant impact on the trajectories. More specifically, directions $\theta_1, \ldots, \theta_n \in \mathcal{S}^{d-1}$ undergo the *normalized attention dynamics* given by

$$\boxed{\dot{\theta}_j(t) = \frac{1}{s_j(t)} \mathbf{P}_{\theta_j(t)} A_j^t(\Theta(t))} \tag{NA}$$

where $\Theta(t) = [\theta_1(t), \ldots, \theta_n(t)]$ and we recall that $\mathbf{P}_\theta = I_d - \theta \theta^\top$ is the projection from $\mathbb{R}^d$ to the tangent space of the sphere at $\theta$. Using the following identities

$$\dot{r}_j(t) = \langle \theta_j(t), \dot{x}_j(t) \rangle,$$

$$\dot{\theta}_j(t) = \frac{1}{r_j(t)} \big(\dot{x}_j(t) - \dot{r}_j(t) \theta_j(t)\big) = \frac{1}{r_j(t)} \mathbf{P}_{\theta_j(t)} \dot{x}_j(t),$$

we readily get:

Table 2: Speed regulation factors

| | $s_j(t)$ | $\dot{r}_j(t)$ |
|---|---|---|
| **Post-LN** | 1 | 0 |
| **Pre-LN** | $r_j(t)$ | $\langle \theta_j(t), A_j^t(\Theta(t)) \rangle$ |
| **Mix-LN** | $\mathbb{1}_{t \leq \tau} + r_j(t)\mathbb{1}_{t > \tau}$ | $\langle \theta_j(t), A_j^t(\Theta(t)) \rangle \mathbb{1}_{t > \tau}$ |
| **Peri-LN** | $r_j(t)\|A_j^t(\Theta(t))\|$ | $\langle \theta_j(t), A_j^t(\Theta(t)) \rangle / \|A_j^t(\Theta(t))\|$ |
| **nGPT** | $\alpha_t^{-1}\|A_j^t(\Theta(t))\|$ | 0 |
| **sqrt-scaling** | $\sqrt{t+1}$ | 0 |

## 3 Asymptotic clustering

Since the work of Geshkovski et al. (2023, 2025), theoretical analyses of attention dynamics have primarily focused on establishing asymptotic clustering under the **Post-LN** scheme, namely $\theta_j(t) \to \theta^*$ as $t \to \infty$ for all $j = 1, \ldots, n$, under a generic initialization; see also Criscitiello et al. (2024); Chen et al. (2025). However, empirical studies have revealed that in practice, tokens often remain trapped in metastable states for extended periods before clustering emerges (Geshkovski et al., 2024a; Bruno et al., 2025a). Despite this, the clustering phenomenon appears to occur at multiple local scales, and the simplified setting considered in prior work continues to offer valuable insights, as we will demonstrate in the next section. In this section, we extend the analysis and show that asymptotic clustering persists beyond the original **Post-LN** framework to other normalization schemes.

Recall that we study the normalized attention dynamics (NA) defined by

$$\dot{\theta}_j(t) = \frac{1}{s_j(t)} \mathbf{P}_{\theta_j(t)} A_j^t(\Theta(t)) = \frac{1}{s_j(t)} \mathbf{P}_{\theta_j(t)} \sum_{k=1}^{n} V\theta_k(t) \frac{e^{\beta\langle Q\theta_j(t), K\theta_k(t)\rangle}}{\sum_{l=1}^{n} e^{\beta\langle Q\theta_j(t), K\theta_l(t)\rangle}} \quad j = 1, \ldots, n,$$

where the speed regulation factor $s_j(t)$ is given in Table 2. It is interesting to note that both **Pre-LN** and **Peri-LN** are not directly regulated by an explicit mechanism but rather by the *magnitude*. In particular, this mechanism dampens the speed of each token individually according to their magnitude.

The main observation of Geshkovski et al. (2025) is that when $KQ^\top = QK^\top = V$, the **Post-LN** system is a gradient flow for the energy function

$$E(\Theta) := -\sum_{j,k=1}^{n} e^{\beta\langle Q\theta_k, K\theta_j\rangle},$$

where we recall that $\Theta = [\theta_1, \ldots, \theta_n]$.

For (NA), we have

$$\dot{\theta}_j(t) = -\frac{1}{s_j(t)Z_j(t)} \nabla_{\theta_j} E(\Theta(t)), \quad \text{where } Z_j(t) = \sum_{l=1}^{n} e^{\beta\langle Q\theta_j(t), K\theta_l(t)\rangle}$$

and $\nabla$ denotes the spherical (Riemannian) gradient.

The above dynamics can be seen as modulated gradient flow, albeit with a complicated modulator that depends on time and space. For vanilla gradient flows, that is for $s_j(t)Z_j(t) = \text{const.}$, a celebrated result of Łojasiewicz guarantees convergence of this gradient flow to a critical point of the energy. Following the same steps, we show in the Appendix D.1 that this result extends to the present framework, guaranteeing convergence of any trajectory. From there, we establish the following clustering result.

**Theorem 3.1.** *Consider the normalized attention dynamics* (NA) *with $Q = K = V = I_d$. Then for uniformly sampled initializations $\Theta(0) \in (\mathcal{S}^{d-1})^{\otimes n}$ Post-LN, nGPT, sqrt-scaling cluster asymptotically*

$$\mathbb{P}[\{\text{tokens synchronize to 1 cluster}\}] = 1,$$

*whereas for a standard Gaussian sample of $X(0) := r(0) \cdot \Theta(0)$ with $\Theta(0) \in (\mathcal{S}^{d-1})^{\otimes n}, r(0) \in \mathbb{R}^{\otimes n}$ for Pre-LN, Mix-LN, Peri-LN one has*

$$\mathbb{P}[\{\text{tokens } \theta_j \text{ synchronize to 1 cluster}\} \cup \{\min_{j \in [n]} \liminf_{t \to \infty} \dot{r}_j(t) = 0\}] = 1.$$

In fact, this result holds not only for $Q^t = K^t = V^t = I_d$ but more generally for $Q^t = Q, K^t = K$, and $V^t = V = Q^\top K = K^\top Q$ as in Sander et al. (2022). The second condition on the magnitude growth can be traced with Table 2 definition to work with further. For example, we immediately get the following.

**Corollary 1.** *For **Pre-LN**, **Peri-LN** with $n \le e^\beta$ we have unconditional synchronization.*

This statement follows from a simple lower bound on $\dot{r}_j$. We write it for **Pre-LN**, and **Peri-LN** can be done similarly.

$$\dot{r}_j = \langle \theta_j, A_j(\Theta) \rangle = \frac{1}{Z_j}(e^\beta \langle \theta_j, \theta_j \rangle + \sum_{k \neq j} e^{\beta \langle \theta_k, \theta_j \rangle} \langle \theta_k, \theta_j \rangle) \geq \frac{1}{ne^\beta}(e^\beta - (n-1)) \geq \frac{1}{ne^\beta},$$

where we used the fact that any negative term in the second sum is at most 1, $e^\beta \geq n$ and a trivial bound on $Z_j$.

## 4 Initial and terminal token velocities

The previous section established an asymptotic result but did not address the rate at which tokens cluster, an aspect that is crucial for understanding how representations evolve. This question is important because the velocity at time $t$ determines the influence of the $t$th layer in shaping the final token representation.

Before analyzing the propagation speed of tokens in our attention dynamics model, we first discuss a benchmark for desirable behavior. In an efficient architecture, each layer should meaningfully transform token representations, causing substantial displacement in representation space. If tokens remain nearly stationary across many layers, the architecture risks *representation collapse*. Equally important, however, is ensuring that early layers contribute significantly—delaying transformation until later stages can limit the expressive power of the network.

### 4.1 Prelude: Symmetric initialization

Following Geshkovski et al. (2025); Cowsik et al. (2024), we begin with a so-called orthogonal symmetric initialization where $\langle \theta_j(0), \theta_k(0) \rangle = 0$ for $j \neq k$ and $r_j(0) = 1$ for all $j$. This configuration approximately matches that of randomly initialized tokens in high dimension. Due to the symmetry, the cosine similarity $\gamma(t) = \langle \theta_j(t), \theta_k(t) \rangle$ does not depend on $j \neq k$ and the entire token dynamics reduces to the evolution of two scalar quantities: $\gamma(t)$ and $r(t)$. In the Appendix, we derive a simple ODE for $\gamma(t), r(t)$ following Geshkovski et al. (2025, Theorem 6.8). We plot ODE-based evolution of $\gamma(t)$ in Figure 2 with parameters $\beta = 5$, $n = 256$. Despite its simplicity, this setup already provides striking insight into the effects of different normalization schemes. The importance tracks in how cosine similarity evolution is alike in the theoretical formula plotted in Figure 2 and the experimental setup with random weights modeled in Figure 4.

**Theorem 4.1.** *Consider the normalized attention dynamics* (NA) *with $Q = K = V = I_d$ initialized at a symmetric orthogonal configuration, i.e. $\langle \theta_j(0), \theta_k(0) \rangle = \delta_{jk}$ and $r_j(0) = r_0$ for all $j$. Then, for all $t > 0$, the cosine similarity $\gamma(t) = \langle \theta_j(t), \theta_k(t) \rangle$ remains constant across all pairs $j \neq k$ and $\dot{\gamma}(t)$ for $t \to 0$ and $t \to \infty$ is given by*

|  | $t \to 0$ | $t \to \infty$ |
|---|---|---|
| **Post-LN** | $\dfrac{2}{e^\beta + n - 1}$ | $Ce^{-2t}$ |
| **Pre-LN** | $\dfrac{2}{r_0(e^\beta + n - 1)}$ | $C/t^3$ |
| **Mix-LN** | $\dfrac{2}{e^\beta + n - 1}$ | $C/t^3$ |
| **Peri-LN** | $\dfrac{2}{r_0\sqrt{e^{2\beta} + n - 1}}$ | $C/t^3$ |
| **nGPT** | $\dfrac{2\alpha_0}{\sqrt{e^{2\beta} + n - 1}}$ | $C\alpha_t e^{-2\int_C^t \alpha_s ds}$ |
| **sqrt-scaling** | $\dfrac{2}{e^\beta + n - 1}$ | $C\dfrac{e^{-4\sqrt{t}}}{\sqrt{t}}$ |

*where $C > 0$ may change from line to line.*

A few remarks are in order. First, the initial velocities are comparable across models, up to the effects of the tuning parameters $\alpha_0$ and $r_0$. Notably, the temperature parameter $\beta$ exponentially damps the initial velocity, suggesting that initializing $Q$ and $K$ with smaller magnitudes in the early layers may be beneficial. More striking is the effect of speed regulation at *terminal velocity*: **Pre-LN**, **Mix-LN**, and **nGPT** (with constant $\alpha_t$) exhibit a polynomial slowdown, in contrast to other normalization schemes. While **sqrt-scaling** converges more slowly than exponential, it still outpaces the polynomial decay. This implies that **Pre-LN**, **Mix-LN**, and **nGPT** cluster more gradually than their counterparts—indicating a more effective use of intermediate layers and a stronger resistance to representation collapse. Finally, note that the trainable parameter $\alpha_t$ in **nGPT** can have a drastic impact on both initial and terminal velocity. See Figure 2 for a visual representation of cosine similarity and evolution of $\dot{\gamma}$ relative to time and position. See Figure 3 for comparison between different $\alpha_t$ in **nGPT**.

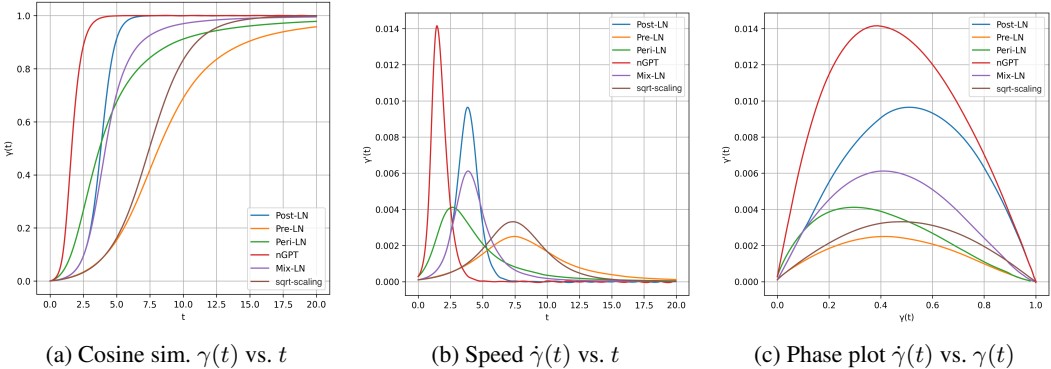

(a) Cosine sim. $\gamma(t)$ vs. $t$      (b) Speed $\dot{\gamma}(t)$ vs. $t$      (c) Phase plot $\dot{\gamma}(t)$ vs. $\gamma(t)$

Figure 2: (a) Evolution of cosine similarity $\gamma(t)$, (b) its speed $\dot{\gamma}(t)$ over time, (c) phase-plot of $\dot{\gamma}(t)$ vs. $\gamma(t)$, for introduced normalization strategies. Here **nGPT** has $\alpha_t \equiv 1$, to showcase the significance of that parameter. **Pre-LN** and **Peri-LN** are the last to converge, mitigating representation collapse. On the other hand, **Post-LN**, **nGPT** and **Peri-LN** move faster in early layers, effectively utilizing them. In the phase-plot (c) we see how at the same position the speed is defined by a known speed control parameter, ranking different methods.

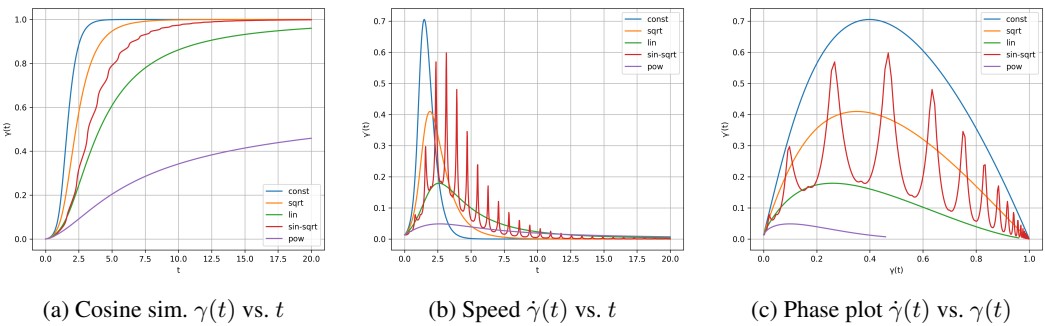

(a) Cosine sim. $\gamma(t)$ vs. $t$      (b) Speed $\dot{\gamma}(t)$ vs. $t$      (c) Phase plot $\dot{\gamma}(t)$ vs. $\gamma(t)$

Figure 3: Convergence in **nGPT** from orthogonal initialization for different choices of $\alpha_t$ – constant, root, linear, combination of linear and constant with weights $\sin(4t)$ and $\cos(4t)$.

## 4.2 Initial velocity

The symmetric evolution described above is too coarse to properly discriminate between normalization schemes at initialization. Here we show that early **Peri-LN/nGPT** layers move tokens *order-one* distances on the hypersphere, while **Post-LN** and **Pre-LN** advance more slowly, with step sizes on the order of $O(\log n/d)$.

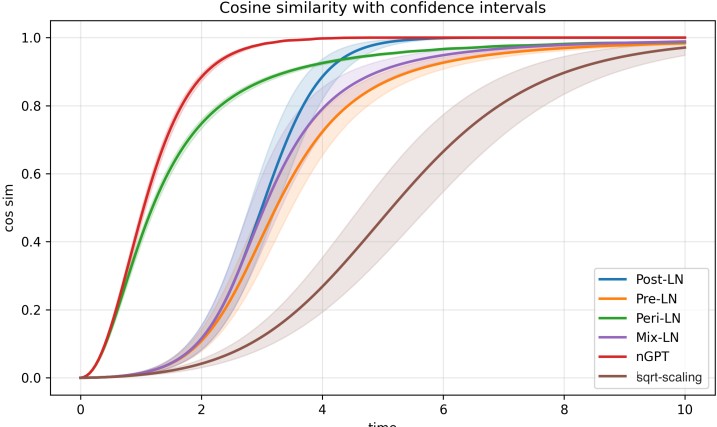

Figure 4: Evolution of average cosine similarity $\gamma(t)$ with 90% confidence interval with randomly initialized weights (Kaiming init), $d = 512, n_{\text{heads}} = 1, \beta = \sqrt{d}, d > n$ and random initial $X$. We set $\alpha_t \equiv 1$ for **nGPT**. We see that **Peri-LN** and **nGPT** initially move faster, and that **Post-LN** and **nGPT** eventually collapse tokens faster than **Pre-LN** and **Peri-LN**. See Appendix E for more studies, including multi-head, untied weights and more.

**Theorem 4.2.** *Let $Q, K, V \in \mathbb{R}^{d \times d}$ satisfy $\max\{\|Q^{\top}K\|_{op}, \|V\|_{op}\} \leq 1$, $\beta = 1$. Let the initial directions $\theta_j(0) \overset{i.i.d.}{\sim} \mathrm{Unif}(\mathbb{S}^{d-1})$ and set the attention vector*

$$A_j(\theta) = \frac{1}{Z_j} \sum_{k=1}^{n} e^{\beta \langle Q\theta_j, K\theta_k \rangle} V\theta_k, \qquad Z_j = \sum_{k=1}^{n} e^{\beta \langle Q\theta_j, K\theta_k \rangle}.$$

*Then there are absolute constants $c, C > 0$ such that for $e^{\sqrt{d}} \geq n \log n \geq d$, with probability $1 - n^{-C}$ simultaneously for all $j \in [n]$*

$$\|A_j(0)\| \leq C \left( \sqrt{\frac{\log n}{n}} + \frac{\log n}{d} \right).$$

To interpret the significance of Theorem 4.2, recall from Table 2 that the initial velocity of direction $\theta_j$ is dampened by a factor proportional to $\|A_j(0)\|$ for both **Peri-LN** and **nGPT**. Consequently the first–layer angular displacement of **Peri-LN** and **nGPT** exceeds that of **Post-LN**, **Pre-LN**, **Mix-LN**, and **sqrt-scaling**, by a factor $\Omega(\min(d/\log n, \sqrt{n/\log n}))$.

### 4.3 Terminal velocity

The idealized setup of Section 4.1 sheds light on a qualitative difference between **Post-LN** and **Pre-LN**: **Post-LN** clusters tokens much more aggressively than **Pre-LN** in the late stages of clustering. In retrospect, the intuition behind this phenomenon is rather clear: under **Pre-LN**, the angular velocity $\theta_j(t)$ of token $j$ is divided by a growing radial factor $r_j(t)$, which increasingly dampens the rate at which tokens collapse toward one another. In contrast, **Post-LN** normalizes this growth away, allowing tokens to continue clustering at a higher rate.

In this section, we go beyond the symmetric case of Section 4.1 and analyze a simplified setting in which tokens are pre-clustered, in the sense that they all lie within a narrow cone. This configuration captures the behavior of a single well-formed cluster and isolates the dynamics from interference by other clusters. The results below confirm our findings of Section 4.1 indicating that this idealized setup is already informative.

**Radial Growth under Pre-LN.** Our first goal is to estimate the rate of growth of $r_j(t)$, the norm of token $j$'s representation, under **Pre-LN** normalization. Empirically, the growth of hidden states in transformers has been well-documented. For instance, studies such as (Xiong et al., 2020; Kedia

et al., 2024) observe that in randomly initialized transformers, $r_j(t) \sim \sqrt{t}$, reflecting the diffusive nature of a random walk induced by randomly sampled projections $V$.

However, in an aligned regime where all tokens are directionally coherent, the dynamics reinforce alignment and exhibit linear radial growth: $r_j(t) \sim t$ as in Section 4.1. This linear scaling significantly alters the clustering behavior. Because the angular update is effectively scaled by $1/r_j(t)$, linear growth in $r_j(t)$ slows the clustering rate from exponential to polynomial.

**Speed of cluster collapse.** To quantify the normalization induced slowdown, we introduce the $\mathrm{Var}(t)$ as a proxy for intra-cluster variance. Specifically, given token directions, $\theta_1(t), \ldots, \theta_n(t)$ let

$$\mathrm{Var}(t) := \frac{1}{n} \sum_{k=1}^{n} \|\theta_k(t) - \bar{\theta}(t)\|^2, \quad \text{where } \bar{\theta} = \frac{1}{n} \sum_{j=1}^{n} \theta_j.$$

**Theorem 4.3.** *Consider the normalized attention dynamics with $V = I_d$ and arbitrary $Q, K$ s.t. $\|Q^\top K\| \leq 1$, initialized at $\theta_1(0), \ldots, \theta_n(0)$ in a local cone, namely $\langle \theta_j(0), \theta_k(0) \rangle \geq 1 - \delta$ for $\delta < 1/(100n^2\beta^2)$. Let the cluster center be defined as $\bar{\theta} = \frac{1}{n} \sum_{j=1}^{n} \theta_j$. Then the following properties hold*

*(i)* **Radial growth** *For all $k$, the radial component satisfies*

$$r_k(t) \geq (1 - \delta)t, \quad \text{for both } \textbf{Pre-LN} \text{ and } \textbf{Peri-LN}$$

*(ii)* **Speed of clustering.** *It holds*

$$\frac{d}{dt}\mathrm{Var}(t) = \begin{cases} -\Theta(\mathrm{Var}(t)), & \text{for } \textbf{Post-LN} \\ -\Theta\left(\mathrm{Var}(t)/t\right) & \text{for } \textbf{Pre-LN} \\ -\Theta\left(\mathrm{Var}(t)/t\right) & \text{for } \textbf{Peri-LN} \\ -\Theta\left(\mathrm{Var}(t)/\alpha_t\right) & \text{for } \textbf{nGPT} \\ -\Theta\left(\mathrm{Var}(t)/t\right) & \text{for } \textbf{Mix-LN} \\ -\Theta\left(\mathrm{Var}(t)/\sqrt{t}\right) & \text{for } \textbf{sqrt-scaling} \end{cases}$$

Again this result corroborates the findings of Section 4.1: **Post-LN** and **sqrt-scaling** cluster token directions at an exponential rate, while **Pre-LN**, **Peri-LN** and **Mix-LN** slow down to a polynomial ($\sim 1/t^C$) decay. Moreover, **nGPT** has the ability to control rate of clustering through $\alpha_t$. This confirms that **Pre-LN makes better use of depth**, as tokens continue to evolve meaningfully across many layers, rather than collapsing too quickly. In particular, this difference explains why **Pre-LN** is less prone to *representation collapse* in very deep models compared to **Post-LN**.

Theorems 4.2–4.3 together give concrete guidelines to select a normalization scheme with large initial and terminal velocities so as to ensure adequate progress of token representations across both first and deep layers. A clear winner here is **Peri-LN** that manages to do both automatically, and **nGPT** that has ability to control the behavior via $\alpha_t$.

## 5  Limitations

Our study offers a unifying dynamical-systems view of several normalization schemes, yet some theoretical and practical caveats temper its scope.

**Theoretical limitations**    Theorem 3.1 proves that every trajectory of the normalized-attention ODE converges. Our approach relies on transforming the system to a compact autonomous frame, therefore it gives some convergence rate(that roughly aligns with Theorem4.1), but it furnishes neither an explicit rate nor any metastability guarantees.

We bound only the initial and terminal speeds; the intermediate regime remains uncharacterized. In particular, one flow could enter a region where it moves faster than another—even though its speed-control factor is larger. Comparing two flows in the general case, even when one enjoys a higher speed-control factor, remains an open problem.

Because of the strict assumptions on the weight matrices, the analysis does not capture the full behavior observed in both theory and practice. For example, in this work representation norms in

Pre-LN are predicted to grow linearly (when matrices Q, K, V are tied), whereas empirical work reports a $\sqrt{t}$ trend at initialization (when weights are random). Reconciling these gaps calls for a stochastic analysis of the problem.

Our theory also leaves unexplained several optimization pathologies—such as exploding updates in Pre-LN—because it omits working with the gradient propagation. A companion gradient-flow analysis is required for a complete picture and is the subject of ongoing work.

**Practical limitations**   From a practical perspective, we make two key simplifications. (i) MLP layers are omitted to focus purely on attention; and (ii) the query, key, and value matrices obey restrictive assumptions. Although the intuition gained from these toy settings is instructive, the proofs rely heavily on the simplifying hypotheses. Finally, in this work, we do not give any specific model architecture to train and validate, which currently limits direct architectural recommendations we could offer.

Addressing these limitations—tight metastability bounds, inclusion of MLP layers, gradient-flow analysis, and empirical verification—constitutes fertile ground for future research.

## Acknowledgments

PR is supported by NSF grants DMS-2022448 and CCF-2106377.

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

# A  Symmetric initialization

This section supplements the results of Section 4.1 by establishing the ODE governing the evolution of cosine similarity $\gamma(t)$ and the magnitude $r(t)$ for each normalization scheme. While Theorem 3.1 guarantees convergence to a point mass from almost all initial configurations, we need to ensure that $\gamma(t) \to 1$ from a symmetric initialization as it approximates a random initial configuration when the embedding dimension $d$ is large. Below, the ODEs governing the evolution of $\gamma(t)$, that is the form of $\dot{\gamma}(t) = 2\langle \dot{\theta}_k(t), \theta_1(t) \rangle$ can be derived using basic substitutions and we omit these details. Moreover, since, **Mix-LN** is simply a combination of **Post-LN** and **Pre-LN**, the initial and terminal velocity in this case follow directly.

**Post-LN.** The ODE governing the evolution of the cosine similarity $\gamma(t)$ was already derived in (Geshkovski et al., 2025, Theorem 6.8). It is given by

$$\dot{\gamma}(t) = \frac{2e^{\beta\gamma(t)}(1 - \gamma(t))((n-1)\gamma(t) + 1)}{((n-1)e^{\beta\gamma(t)} + e^{\beta})} \, .$$

At $t = 0$, $\gamma(t) = 0$ and it is known from the aforementioned theorem that $\gamma(t) \to 1$ as $t \to \infty$. In fact we readily see from the ODE that $\gamma(t)$ is monotonically increasing. Writing $\varepsilon(t) = 1 - \gamma(t)$, we get $\dot{\varepsilon}(t) \sim -2\varepsilon(t)$. It yields

$$\dot{\gamma}(t) \sim_{t \to 0} \frac{2}{e^{\beta} + n - 1} \, , \qquad \dot{\gamma}(t) \sim_{t \to \infty} Ce^{-2t}.$$

**Pre-LN.** The ODEs governing $r(t)$ and $\gamma(t)$ are given by

$$\dot{r}(t) = \frac{(n-1)e^{\beta\gamma(t)}\gamma(t) + e^{\beta}}{(n-1)e^{\beta\gamma(t)} + e^{\beta}}$$

and

$$\dot{\gamma}(t) = \frac{2e^{\beta\gamma(t)}(1 - \gamma(t))((n-1)\gamma(t) + 1)}{r(t)((n-1)e^{\beta\gamma(t)} + e^{\beta})} \, . \tag{1}$$

Note that $\gamma$ is increasing so $\gamma(t) \geq \gamma(0) = 0$ for all positive $t$. Hence,

$$\dot{\gamma}(t) \geq \frac{2e}{r(t)ne^{\beta}}(1 - \gamma(t)) \, .$$

By Grönwall's inequality, we get

$$1 - \gamma(t) \leq \exp\left(-\frac{2e}{ne^{\beta}} \int_0^t \frac{ds}{r(s)}\right)$$

But since $\dot{r} \leq 1$, we have $r(t) \leq t + r(0)$ and $\int_0^t \frac{ds}{r(s)} \to \infty$ as $t \to \infty$. Hence $\gamma(t) \to 1$ and, in turn, $\dot{r}(t) \to 1$ so that[3] $r(t) \sim_{t \to \infty} t$ as $t \to \infty$ by l'Hôpital's rule.

Writing $\varepsilon(t) = 1 - \gamma(t)$, we get $\dot{\varepsilon}(t) \sim_{t \to \infty} -2\varepsilon(t)/r(t) \sim_{t \to \infty} -2\varepsilon(t)/t$. It yields that

$$\dot{\gamma}(t) \sim_{t \to 0} \frac{2}{r(0)(e^{\beta} + n - 1)} \, , \qquad \dot{\gamma}(t) \sim_{t \to \infty} \frac{C}{t^3}.$$

**Peri-LN.** The ODEs governing $r(t)$ and $\gamma(t)$ are given by

$$\dot{r}(t) = \frac{(n-1)e^{\beta\gamma(t)}\gamma(t) + e^{\beta}}{\sqrt{e^{2\beta} + 2(n-1)e^{\beta(1+\gamma(t))}\gamma(t) + (n-1)e^{2\beta\gamma(t)}(1 + (n-2)\gamma(t))}}$$

and

$$\dot{\gamma}(t) = \frac{2e^{\beta\gamma(t)}(1 - \gamma(t))((n-1)\gamma(t) + 1)}{r(t)\sqrt{e^{2\beta} + 2(n-1)e^{\beta(1+\gamma(t))}\gamma(t) + (n-1)e^{2\beta\gamma(t)}(1 + (n-2)\gamma(t))}}$$

---

[3]For two function $a(t)$ and $b(t)$ and $T \in \{0, \infty\}$, we write $a(t) \sim_{t \to T} b(t)$ if $a(t)/b(t) \to 1$ as $t \to T$.

The argument follows the same lines as for **Pre-LN**. Indeed, we have

$$\dot{\gamma}(t) \geq \frac{2e}{r(t)e^\beta \sqrt{1 + (n-1)^2}}(1 - \gamma(t)) \geq \frac{2e}{r(t)ne^\beta}(1 - \gamma(t)),$$

and hence

$$1 - \gamma(t) \leq \exp\left(-\frac{2e}{ne^\beta}\int_0^t \frac{\mathrm{d}s}{r(s)}\right).$$

To show that $\dot{r} \leq 1$ in this case too, we employ a coarser approximation that is sufficient for our purpose:

$$\dot{r}(t) \leq \frac{ne^\beta}{\sqrt{e^{2\beta} + n - 1}} \leq n.$$

It readily yields that $\gamma(t) \to 1$ as $t \to \infty$ and in turn that $r(t) \sim_{t\to\infty} t$. Hence,

$$\dot{\gamma}(t) \sim_{t\to 0} \frac{2}{r(0)\sqrt{e^{2\beta} + n - 1}}, \qquad \dot{\gamma}(t) \sim_{t\to\infty} \frac{C}{t^3}.$$

**nGPT.** The ODE governing $\gamma(t)$ is given by

$$\dot{\gamma}(t) = \frac{2\alpha_t e^{\beta\gamma(t)}(1 - \gamma(t))((n-1)\gamma(t) + 1)}{\sqrt{e^{2\beta} + 2(n-1)e^{\beta(1+\gamma(t))}\gamma(t) + (n-1)e^{2\beta\gamma(t)}(1 + (n-2)\gamma(t))}}, .$$

This is the same formula as **Peri-LN** where $r(t)$ is replaced with $\alpha_t^{-1}$. Hence,

$$1 - \gamma(t) \leq \exp\left(-\frac{2e}{ne^\beta}\int_0^t \alpha_s \mathrm{d}s\right)$$

Assuming that $\alpha_t$ is chosen such that the above integral diverges as $t \to \infty$, we get that $\gamma(t) \to 1$ as $t \to \infty$. It yields

$$\dot{\gamma}(t) \sim_{t\to 0} \frac{2\alpha_0}{\sqrt{e^{2\beta} + n - 1}}, \qquad \dot{\gamma}(t) \sim_{t\to\infty} C\alpha_t e^{-2\int_0^t \alpha_s \mathrm{d}s}.$$

**sqrt-scaling.** The ODE governing $\gamma(t)$ is given by

$$\dot{\gamma}(t) = \frac{2e^{\beta\gamma(t)}(1 - \gamma(t))((n-1)\gamma(t) + 1)}{\sqrt{t+1}((n-1)e^{\beta\gamma(t)} + e^\beta)}$$

Observe that the cosine similarity evolves precisely as (1) but with predetermined magnitude $r(t) = \sqrt{t+1}$. In particular, we get that $\gamma(t) \to 1$ as $t \to \infty$. We readily get

$$\dot{\gamma}(t) \sim_{t\to 0} \frac{2}{e^\beta + n - 1}, \qquad \dot{\gamma}(t) \sim_{t\to\infty} C\frac{e^{-4\sqrt{t}}}{\sqrt{t}}.$$

# B   Proof of Theorem 4.2

Here we prove an upper bound on the initial attention vector. Assume $\beta = 1$, $n \log n \geq d \geq \log^2 n$, $\|Q^\top K\|_{\mathrm{op}}, \|V\|_{\mathrm{op}} \leq 1$, i.i.d. random uniform $\theta_j$. Then

$$\mathbb{P}\left(\forall j \in [n] \; \|A_j(\Theta(0))\| \leq C\frac{\log n}{d}\right) \geq 1 - n^{-C}.$$

*Proof.* Throughout this proof, $C > 0$ denotes a universal constant that may change from line to line.

Fix token $j$—without loss of generality, $j = n$—and work conditionally on $\theta_n$. Define the random variables:

$$X_k := \theta_n^\top Q^\top K\theta_k, \quad k = 1, \ldots, n.$$

Our goal is to control the norm of the vector

$$A_n(\Theta(0)) := V\frac{\sum_{k=1}^n e^{X_k}\theta_k}{\sum_{k=1}^n e^{X_k}}.$$

Since we assume that $\|V\|_{op} \leq 1$, we may assume without loss of generality that $V = I_d$.

Let $w$ denote the probability vector given by $w_k \propto e^{X_k}$ and observe that

$$\sum_{k=1}^{n} w_k \theta_k = \frac{1}{n} \sum_{k=1}^{n} \theta_k + \frac{1}{n} \sum_{k=1}^{n} X_k \theta_k + \sum_{k=1}^{n} \left(w_k - \frac{1}{n} - \frac{X_k}{n}\right)\theta_k. \tag{2}$$

Since the $\theta_k$s are i.i.d. centered and subGaussian with variance proxy $C/d$, we get that with probability at least $1 - n^{-C}$

$$\|\frac{1}{n} \sum_{k=1}^{n} \theta_k\| \leq C\sqrt{\frac{\log n}{n}} \tag{3}$$

Moreover, observe that for any $k \leq n$, we have

$$\|\mathbb{E}\frac{1}{n} \sum_{k=1}^{n} X_k \theta_k\| \leq \frac{C}{d}.$$

Hence, by vector Hoeffding, with probability at least $1 - n^{-C}$, we also have

$$\|\frac{1}{n} \sum_{k=1}^{n} X_k \theta_k\| \leq \frac{C}{d} + C\sqrt{\frac{\log n}{n}} .$$

because we assumed $n \log n \geq d$.

We now control the third and last term in the right-hand side of (2). and observe that $X_n$ is deterministic and with norm at most 1. For $k \leq n - 1$, the random variables $X_k$ are i.i.d centered and subGaussian with variance proxy $C/d$. Hence there exists an event $E$, with probability at least $1 - n^{-C}$, on which

$$\max_{k \leq n-1} |X_k| \leq C\sqrt{\frac{\log n}{d}} .$$

Since $n \leq e^{\sqrt{d}}$, on $E$, it holds for all $k \leq n - 1$,

$$|e^{X_k} - 1 - X_k| \leq C\frac{\log n}{d} .$$

Moreover, we have that $|X_n| \leq 1$ so that $e^{-1} \leq e^{X_n} \leq e$. Together, these bounds yield that

$$\frac{1 + X_k - C\frac{\log n}{d}}{n - 1 + e} \leq w_k \leq \frac{1 + X_k + C\frac{\log n}{d}}{n - 1 + e^{-1}} , \quad k \leq n - 1 ,$$

so that

$$\left|w_k - \frac{1}{n} - \frac{X_k}{n}\right| \leq C\frac{\log n}{nd}$$

where we used the fact that $n \geq \sqrt{d}$. Moreover, using similar arguments, we also have

$$\left|w_k - \frac{1}{n} - \frac{X_k}{n}\right| \leq 2 .$$

Put together, the last two displays yield

$$\left\|\sum_{k=1}^{n} \left(w_k - \frac{1}{n} - \frac{X_k}{n}\right)\theta_k\right\| \leq C\frac{\log n}{d} .$$

Combined together we get the claimed estimate. $\square$

## C   Proof of Theorem 4.3

Denote average $\bar{\theta} = \frac{1}{n} \sum_{k=1}^{n} \theta_k$. Consider variance of tokens

$$\mathcal{V}(t) := \frac{1}{n} \sum_{k=1}^{n} \|\theta_k - \bar{\theta}\|^2 = 1 - \|\bar{\theta}\|^2.$$

Then
$$\mathcal{V}'(t) = \frac{2}{n}\sum_{k=1}^{n}\langle\theta_k - \bar\theta, \dot\theta_k - \frac{1}{n}\sum_{j=1}^{n}\dot\theta_j\rangle.$$

We immediately have
$$\sum_{k=1}^{n}\langle\theta_k - \bar\theta, \frac{1}{n}\sum_{j=1}^{n}\dot\theta_j\rangle = \langle\sum_{k=1}^{n}\theta_k - n\bar\theta, \frac{1}{n}\sum_{j=1}^{n}\dot\theta_j\rangle = 0.$$

Thus
$$\mathcal{V}'(t) = \frac{2}{n}\sum_{k=1}^{n}\langle\theta_k - \bar\theta, \dot\theta_k\rangle = \frac{2}{n}\sum_{k=1}^{n}\langle\theta_k - \bar\theta, \frac{1}{s_k}P_k A_k\rangle.$$

Let's decompose $\delta_k := A_k - \theta_k$ to get
$$\mathcal{V}'(t) = \frac{2}{n}\sum_{k=1}^{n}\langle\theta_k - \bar\theta, \frac{1}{s_k}P_k\bar\theta\rangle + \frac{2}{n}\sum_{k=1}^{n}\langle\theta_k - \bar\theta, \frac{1}{s_k}P_k\delta_k\rangle = I_1 + I_2.$$

For the first term we write
$$\frac{n}{2}I_1 = \sum_{k=1}^{n}\frac{1}{s_k}\langle\theta_k - \bar\theta, P_k\bar\theta\rangle = \sum_{k=1}^{n}\frac{1}{s_k}\langle-\bar\theta, \bar\theta - \langle\theta_k, \bar\theta\rangle\theta_k\rangle = \sum_{k=1}^{n}\frac{1}{s_k}(\langle\theta_k, \bar\theta\rangle^2 - \|\bar\theta\|^2).$$

Each term in the sum is non-positive, thus we can bound
$$\frac{1}{\max_k s_k}\sum_{k=1}^{n}(\langle\theta_k, \bar\theta\rangle^2 - \|\bar\theta\|^2) \geq \frac{n}{2}I_1 \geq \frac{1}{\min_k s_k}\sum_{k=1}^{n}(\langle\theta_k, \bar\theta\rangle^2 - \|\bar\theta\|^2).$$

The sum itself can be written as
$$\sum_{k=1}^{n}(\langle\theta_k, \bar\theta\rangle^2 - \|\bar\theta\|^2) = \sum_{k=1}^{n}\langle\theta_k, \bar\theta\rangle^2 - n\|\bar\theta\|^2 = \sum_{k=1}^{n}(\langle\theta_k, \bar\theta\rangle^2 - \langle\theta_k, \bar\theta\rangle),$$

since $\sum_{k=1}^{n}\langle\theta_k, \bar\theta\rangle = n\|\bar\theta\|^2 = n - n\mathcal{V}(t)$. With a fixed sum of $\langle\theta_k, \bar\theta\rangle$, the min/max sum of squares $\langle\theta_k, \bar\theta\rangle^2$ is achieved when they are equal/spread out, which gives us
$$\frac{1}{\max_k s_k}(-2\mathcal{V} + 2n\mathcal{V}^2) \geq I_1 \geq \frac{1}{\min_k s_k}(-2\mathcal{V} + 2\mathcal{V}^2).$$

For the second term, we first upper bound the length of $P_k\delta_k$. To this aim, consider
$$\langle Q\theta_k, K\theta_i\rangle - \langle Q\theta_k, K\theta_j\rangle = \langle\theta_k, Q^\top K(\theta_i - \theta_j)\rangle \leq \|\theta_k\|\|Q^\top K\|_{op}\|\theta_i - \theta_j\| \leq \|\theta_i - \theta_j\| \leq \sqrt{2\delta}.$$

Consequently,
$$\frac{1}{ne^{-\beta\sqrt{2\delta}}} \geq \frac{e^{\beta\langle Q\theta_k, K\theta_j\rangle}}{\sum_{t=1}^{n}e^{\beta\langle Q\theta_k, K\theta_t\rangle}} \geq \frac{1}{ne^{\beta\sqrt{2\delta}}}.$$

Which implies
$$\left|w_{kj} - \frac{1}{n}\right| = \left|\frac{e^{\beta\langle Q\theta_k, K\theta_j\rangle}}{\sum_{t=1}^{n}e^{\beta\langle Q\theta_k, K\theta_t\rangle}} - \frac{1}{n}\right| \leq \frac{1}{n}(e^{\beta\sqrt{2\delta}} - 1).$$

Therefore,
$$\|P_k\delta_k\| = \|\sum_{j=1}^{n}(w_{kj} - \frac{1}{n})P_k\theta_j\| \leq \frac{1}{n}(e^{\beta\sqrt{2\delta}} - 1)\sum_{j=1}^{n}\|P_k\theta_j\| \leq \frac{1}{n}(e^{\beta\sqrt{2\delta}} - 1)\sqrt{n}\sqrt{\sum_j\|P_k\theta_j\|^2}.$$

Finally, one has
$$\sum_{j=1}^{n}\|P_k\theta_j\|^2 = \sum_{j=1}^{n}(1 - \langle\theta_j, \theta_k\rangle^2) \leq n - \frac{1}{n}(\sum_{j=1}^{n}\langle\theta_j, \theta_k\rangle)^2$$
$$= n(1 - \langle\bar\theta, \theta_k\rangle^2) \leq n(1 - (1 - n\mathcal{V})^2) \leq 2n^2\mathcal{V}.$$

Combined, we obtain an upper bound

$$|I_2| \leq \frac{2}{n} \sum_{k=1}^n \frac{1}{s_k} \|\theta_k - \bar\theta\| \|P_k \delta_k\| \leq \frac{2}{n} \frac{1}{\min_k s_k} \frac{1}{n} (e^{\beta\sqrt{2\delta}} - 1)\sqrt{n}\sqrt{2n^2\mathcal{V}} \sum_{k=1}^n \|\theta_k - \bar\theta\|$$

$$\leq 2\frac{1}{\min_k s_k}(e^{\beta\sqrt{2\delta}} - 1)\sqrt{2\mathcal{V}}(\sum_{k=1}^n \|\theta_k - \bar\theta\|^2)^{1/2} = 2\frac{1}{\min_k s_k}\sqrt{2n}(e^{\beta\sqrt{2\delta}} - 1)\mathcal{V}.$$

Thus, we obtain upper and lower bounds on $\mathcal{V}'(t) = I_1 + I_2$ in terms of $\mathcal{V}$.

$$\frac{-2\mathcal{V} + 2n\mathcal{V}^2}{\max_k s_k} + \frac{2\sqrt{2n}(e^{\beta\sqrt{2\delta}} - 1)}{\min_k s_K}\mathcal{V} \geq \mathcal{V}'(t) \geq \frac{-2\mathcal{V} + 2\mathcal{V}^2 - 2\sqrt{2n}(e^{\beta\sqrt{2\delta}} - 1)}{\min_k s_k}\mathcal{V}. \quad (4)$$

Let us also mention that

$$2\delta = \max_{k,j} \|\theta_k - \theta_j\|^2 \leq 4\max_k \|\theta_k - \bar\theta\|^2 \leq 4n\mathcal{V},$$

whereas

$$1 - \mathcal{V} = \langle\bar\theta, \bar\theta\rangle \geq 1 - \delta, \text{ i.e. } \mathcal{V} \leq \delta.$$

Therefore, the true local rate of clustering that we get from bounds (4) is defined by the main terms on both sides $-2\mathcal{V}/\max_k s_k$ and $-2\mathcal{V}/\min_k s_k$. Moreover, as $\mathcal{V} \to 0$, $\min s_k \sim \max s_k$, so we obtain a tight rate of convergence. To establish the result we claimed, notice that for $\delta < \frac{1}{100n^2\beta^2}$ one has

$$2\sqrt{2n}(e^{\beta\sqrt{2\delta}} - 1) \leq \frac{\sqrt{2}}{3\sqrt{n}}, \qquad 2n\mathcal{V}^2 \leq 2n\delta\mathcal{V},$$

giving us

$$\frac{-2 + 2n\delta + \sqrt{2}/(3\sqrt{n})}{\max_k s_k}\mathcal{V} \geq \mathcal{V}' \geq \frac{-2 - \sqrt{2}/(3\sqrt{n})}{\min_k s_k}\mathcal{V}. \quad (5)$$

Finally, we finish the proof with trivial estimates on $s_k$, that follow from the fact that all products $\langle\theta_k, \theta_j\rangle \geq 1 - \delta$ and definitions.

- For **Post-LN** $s_k \equiv 1$.
- for **Pre-LN** $t \geq s_k \geq (1 - \delta)t$.
- for **Peri-LN** $t \geq s_k \geq (1 - \delta)^{3/2}t$
- for **nGPT** $\alpha_t \geq s_k \geq (1 - \delta)^{1/2}\alpha_t$
- for **Mix-LN** $t \geq s_k \geq (1 - \delta)(t - \tau)$
- for **sqrt-scaling** $s_k \equiv \sqrt{t}$.

Substituted into the estimate (5), we obtain the claimed rates.

**Remark 1.** *The true local rate of convergence of $\mathcal{V}$ as $t \to \infty$ that we get from equation (4) is*

- $\mathcal{V} = e^{-2t(1+o(1))}$ *for **Post-LN**,*

- $\mathcal{V} = e^{-2\log t(1+o(1))}$ *for **Pre-LN**,*

- $\mathcal{V} = e^{-2\log t(1+o(1))}$ *for **Peri-LN**,*

- $\mathcal{V} = e^{-2\int_0^t \alpha_s ds(1+o(1))}$ *for **nGPT**,*

- $\mathcal{V} = e^{-2\log t(1+o(1))}$ *for **Mix-LN**,*

- $\mathcal{V} = e^{-4\sqrt{t}(1+o(1))}$ *for **sqrt-scaling**.*

# D   Final convergence

In this section we prove Theorem 3.1 from the main text, that claims that under some assumptions, for almost any initial configuration of particles, any normalized attention dynamics that we study (that is **Post-LN**, **Pre-LN**, **Peri-LN**, **nGPT**, **Mix-LN** and **sqrt-scaling**) converges to a single cluster. First, let us outline the core of the proof.

## D.1 Proof Outline for Token Synchronization in Pre-LN

A conventional proof that all tokens converge to a single state consists of two stages. Showing that there is some limiting configuration of tokens, and then verifying that the only possible limiting configuration is the consensual one. We follow the same approach, but at each step we introduce novel technical details due to our general point of view. For simplicity of exposition, in the outline we follow **Pre-LN** case.

**Existence of a Limit Point** First, we demonstrate that the token dynamics indeed converge to a limiting configuration. This step heavily depends on the system. Common approach leverages the Łojasiewicz inequality, as seen in Geshkovski et al. (2025). It can be adopted to our setting, as we will show later. Moreover, our proof extends the gradient case $Q^\top K = V$ to a more general case, extending the synchronization results by Geshkovski et al. (2025), Criscitiello et al. (2024), even in **Post-LN** case.

**Local behavior at the limiting point.** Second, we must prove that any such limit point corresponds to the synchronized state where all tokens are identical. The classical argument involves a local stability analysis around the system's critical points. One can typically show that any non-synchronized critical points are unstable and that their basin of attraction has measure zero, making them insignificant as final states. A comprehensive linearization analysis can be found in Criscitiello et al. (2024) that, in particular, covers **Post-LN** dynamics with $d \geq 3$. Together with a recent proof of synchronization for $d = 2$ Polyanskiy et al. (2025), the stability of **Post-LN** system Jacobian is well-studied. We also rely on this method, but first we need to resolve the fact that **Pre-LN** system is non-compact.

**Transformation to compact state space.** The **Pre-LN** state-space is non-compact, because both empirically and theoretically tokens' magnitude $r_j$ grows to infinity with $t$. This restricts the direct study of the limiting point in that space. We can transform it to a compact state space by the following trick, however. Consider a logarithmic time scale $\tau := \ln t$ and modified scale variables $q_j := s_j/t$. Applying the chain rule, we find the transformed dynamics:

$$\frac{d\theta_j}{d\tau} = \frac{1}{q_j} P_j A_j(\Theta)$$

$$\frac{dq_j}{d\tau} = \langle \theta_j, A_j(\Theta) \rangle - q_j.$$

This formulation is interesting in its own right. It reveals that the **Pre-LN** system evolves on a logarithmic time scale, which may explain its observed stability advantages over **Post-LN** variants in deep architectures. Furthermore, the dynamics are scaled by $q_j$, which are driven toward $\langle \theta_j, A_j(\Theta) \rangle$, the alignment between a token and its attention vector.

Crucially for our proof, this transformed system is still *autonomous*. This allows us to proceed with the final step: a rigorous linearization analysis of its critical points. By showing that all critical points corresponding to non-consensual states are unstable in the $(\theta, q, \tau)$ frame, we can conclude that the system must converge to the state where all tokens are identical.

In what follows we are going to cover all the proof steps in detail.

## D.2 Generalized gradient descent convergence

First, we need to refine an important result of Łojasiewicz on convergence of gradient descent, so that it fits our problem setting. We follow an approach similar to the one presented in Haraux (2012).

**Lemma 1.** *For any $t \geq 0$, let $M(t)$ be a symmetric real matrix $C\lambda(t)I \succ M(t) \succ \lambda(t)I$ with $\lambda(t) > 0$, $\int_0^\infty \lambda(t)dt = \infty$, and some constant $C$. Let energy function $E(x)$ be analytic in an open set $U \subset \mathbb{R}^N$. Consider a compact path $x(t) \subset U$ that satisfies the following modified gradient descent equation*

$$\dot{x} = -M(t)\nabla_x E(x).$$

*Then, $x(t)$ converges to a critical point of the energy function $x(t) \to x^*$ such that $\nabla E(x^*) = 0$.*

*Proof. Step I. Change of time.* First, define a new time variable $\tau(t) = \int_0^t \lambda(s)ds$. By assumption $\tau$ monotonically grows to infinity as $t \to \infty$. Moreover,

$$\frac{dx}{d\tau} = \frac{dx/dt}{d\tau/dt} = \frac{-M(t)\nabla_x E(x)}{\lambda(t)}.$$

Take $\tilde{M}(\tau) = M(t(\tau))/\lambda(t(\tau))$. Then

$$\frac{dx}{d\tau} = -\tilde{M}(\tau)\nabla_x E(x)$$

with $CI \succ \tilde{M}(\tau) \succ I$. This change of time proves that it is sufficient to prove the Lemma in its initial form under the assumption $CI \succ M(t) \succ I$, whereas $\lambda(t)$ corresponds to time change.

*Step 2.* Now that we have $CI \succ M(t) \succ I$, let us follow a known approach to the proof of gradient descent convergence. Consider the energy along the trajectory, i.e.

$$f(t) := E(x(t)).$$

Then

$$f'(t) = (\dot{x})^\top \nabla_x E|_{x(t)} = -(\dot{x})^\top M^{-1}\dot{x} \leq -C^{-1}|\dot{x}|^2.$$

In particular, $f'(t) < 0$, the energy is decreasing along the trajectory. Since $E$ is bounded on a compact trajectory, we get that $f'(t) \in L_1([0,\infty))$. Because

$$|\dot{x}|^2 \leq C|f'(t)|,$$

we get that $\dot{x} \in L_2([0,\infty))$. This implies that $\dot{x} \to 0$, because $\dot{x}$ is an absolutely continuous function in $L_2([0,\infty))$.

Therefore, since $M(t) \succ cI$, we get that $\nabla_x E(x) \to 0$. For convergence to a point this is not enough, but it already shows us that $\text{dist}(x, \mathcal{E}) \to 0$ where $\mathcal{E} = \{a : \nabla E(a) = 0\}$. Then, because the limit set $\Gamma$ of a compact trajectory $x(t)$ is compact and connected, we can use uniform Łojasiewicz inequality.

To get $x \to x^*$ we need to sharpen the estimate on $\dot{x}$. This is where the Łojasiewicz inequality is used. It says that in some neighbourhood $\Omega$ of $\Gamma$ and some constants $V, \alpha$ one has

$$|E(u) - V|^\alpha \leq \|\nabla E(u)\|.$$

We can assume $V = 0$ by shifting the energy function. In particular, it means that $f(t)$ decreases to $0$ as $t \to \infty$. Moreover, because $x(t)$ approaches $\Gamma$ as $t \to \infty$, we know that as $t \to \infty$ it is true that

$$|E(x(t))|^\alpha \leq \|\nabla E(x(t))\|.$$

Therefore, from our assumption $M(t) \succ I$ we get

$$f'(t) = (\nabla_x E|_{x(t)})^\top \dot{x} = -(\nabla_x E|_{x(t)})^\top M(t)\nabla_x E|_{x(t)} \leq -\|\nabla_x E(x(t))\|^2 \leq -|f(t)|^{2\alpha}.$$

Then

$$(f^{1-2\alpha}(t))' = (1 - 2\alpha)f^{-2\alpha}f' \geq (2\alpha - 1).$$

Consequently, for $\beta = 1/(2\alpha - 1)$ one has

$$f(t) \leq Kt^{-\beta}.$$

We know that

$$|\dot{x}|^2 \leq C|f'(t)| = -Cf'(t).$$

Then

$$\int_t^{2t} |\dot{x}|^2 ds \leq C(f(t) - f(2t)) \leq CKt^{-\beta}.$$

From this inequality and Cauchy-Schwarz we get

$$\int_t^{2t} |\dot{x}| ds \leq CKt^{(1-\beta)/2}.$$

Finally, this estimate shows convergence of the path $x(t)$ to some limiting point, because

$$\int_1^\infty |\dot{x}| \leq CK \sum_{n=0}^\infty 2^{n(1-\beta)/2} < \infty.$$

$\square$

## D.3 Proof of Theorem 3.1

In this appendix we provide a complete proof of Theorem 3.1. The argument follows the roadmap outlined in Section D.1, with minor adjustments for each normalization scheme. For simplicity of exposition, we first give a full analysis of **Pre-LN**. Then, we provide remarks on how to adapt the proof for each normalization scheme. Thanks to our unified formulation of normalization in (NA), the core proof applies verbatim across all schemes—the only variation lies in some technical details. A forthcoming work will pursue that broader unification and extend the analysis beyond purely architectural speed regulators.

For convenience, let us recall the object of study. We consider the evolution of particles $\theta_j$ on a unit sphere $\mathbb{S}^{d-1}$ governed by the ODE

$$\dot{\theta}_j = \frac{1}{s_j} P_j A_j, \qquad A_j = \sum_{k=1}^{n} \frac{e^{\beta \langle Q\theta_j, K\theta_k \rangle}}{\sum_{\ell=1}^{n} e^{\beta \langle Q\theta_j, K\theta_e ll \rangle}} V\theta_k$$

with normalization factor $s_k$ evolving according to the following table.

Table 3: Speed regulation factors

|  | $s_j(t)$ | $\dot{r}_j(t)$ |
|---|---|---|
| **Post-LN** | $1$ | $0$ |
| **Pre-LN** | $r_j(t)$ | $\langle \theta_j(t), A_j^t(\Theta(t)) \rangle$ |
| **Mix-LN** | $\mathbb{I}_{t \leq \tau} + r_j(t)\mathbb{I}_{t > \tau}$ | $\langle \theta_j(t), A_j^t(\Theta(t)) \rangle \mathbb{I}_{t > \tau}$ |
| **Peri-LN** | $r_j(t)\|A_j^t(\Theta(t))\|$ | $\langle \theta_j(t), A^t(\theta_j(t)) \rangle / \|A_j^t(\Theta(t))\|$ |
| **nGPT** | $\alpha_t^{-1}\|A_j^t(\Theta(t))\|$ | $0$ |
| **sqrt-scaling** | $\sqrt{t+1}$ | $0$ |

**Proposition 1.** *Consider monotonically growing to infinity time change $\tau(t)$. Then, normalized attention dynamics with speed regulation factors $s_j(t)$ is equivalent to normalized attention dynamics with speed regulation factors $\tilde{s}_j(\tau) = s_j(t(\tau))/t'(\tau)$ in time $\tau$.*

*Proof.* This immediately follows from the definition

$$\frac{d\theta_j}{d\tau} = \frac{d\theta_j}{dt} t'(\tau) = \frac{1}{s_j(t(\tau))/t'(\tau)} P_j A_j(\Theta).$$

$\square$

This proposition shows that in the normalized attention dynamics we can divide $s_j$ by the same factor, as long as it's positive and its inverse integrates to infinity. This notion helps us reduce time dependence in normalization dynamics.

*Proof. Step 1. Time change*

Consider evolution starting at time $t = 1$ and a time change $\tau := \ln t$ so that $dt/d\tau = t$. Moreover, set $q_j(t) := r_j(t)/t$. Then, we rewrite **Pre-LN** in time $\tau$ as

$$\dot{\theta}_j(\tau) = \frac{1}{q_j(\tau)} P_j A_j(\Theta(\tau)), \qquad \dot{q}_j(\tau) = \langle \theta_j(\tau), A_j(\Theta(\tau)) \rangle - q_j(\tau).$$

The function $\langle \theta_j, A_j(\Theta) \rangle$ is continuous and thus bounded on the compact. Then, all $q_j$ are upper bounded from the equation, and thus evolve on a segment $[0, Q]$. This frame change is important, as it allows us to study an autonomous system on a compact, whereas in the original coordinates one usually has $r_j \to \infty$. Moreover, the condition

$$\inf_j \liminf_{t \to \infty} \dot{r}_j > 0$$

implies that all magnitudes $r_j$ are lower bounded by some linear function at $t \to \infty$, which translates into

$$\inf_j \inf_\tau q_j(\tau) > 0.$$

*Step 2. Gradient-like structure.* We consider the event $\{\inf_j \inf_\tau q_j > 0\}$. It is enough to show synchronization under this assumption to prove the result. First, to show the convergence of the system to some limiting configuration of angles $\Theta^*$, we use Lemma 1. For any trajectory $\Theta(\tau)$ we can write

$$\dot{\theta}_j = -\frac{1}{q_j Z_j} \nabla_{\theta_j} E(\Theta)$$

with spherical gradient of the following energy function

$$E(\Theta) = -\frac{1}{2\beta} \sum_{j,k} e^{\beta \langle \theta_j, \theta_k \rangle}.$$

To get convergence of a specific trajectory $\Theta(\tau)$ to some critical point $\Theta^*$, we need to verify that the time-dependent matrix $M(\tau)$ with diagonal blocks $\frac{1}{q_j Z_j}$ satisfies the assumptions of Lemma 1. This is true, because the blocks are uniformly bounded. Indeed, the function $Z_j$ is uniformly bounded as continuous functions on a compact. Whereas $q_j$ are uniformly bounded on any trajectory we consider, with $\{\inf_j \inf_\tau q_j > 0\}$.

*Step 3. Local behavior* We consider the event $\inf_j \inf_\tau q_j > 0$ and $\Theta(\tau) \to \Theta^*$. Our goal is to show that when the limiting point is not $\theta_1^* = \ldots = \theta_n^*$, this event has probability zero. We can split the event into a countable union with assumptions $\{q_j(\tau) \geq \frac{1}{m}\}$.

$$\{\inf_j \inf q_j > 0\} \subset \bigcup_{m \in \mathbb{Z}_{>0}} \{\forall j \in [n] \; \forall \tau > 0 \; q_j \geq \frac{1}{m}\}.$$

As we already mentioned, $q_j$ are bounded from above. This means that under the restriction $q_j \geq \frac{1}{m}$, the combined state space of $(\Theta, q)$ is a compact manifold. Our goal is to show that the event

$$\{\forall j \in [n] \; \forall t > 0 \; q_j > \frac{1}{m}\} \cup \{\forall j \in [n] \; \theta_j \to \theta_j^* \mid \Theta^* \text{is not synchronized}\}$$

has probability zero.

When we get an autonomous dynamical system on a compact manifold, and we study its convergence to a limiting point, we need to study the Jacobian at that limiting point. Specifically, a well-known stability argument that was already written down several times (see Criscitiello et al. (2024), (Geshkovski et al., 2025, Lemma A.1)), employs central manifold theorem to show that basin of attraction of unstable critical points has measure zero.

This argument applies to our case. Therefore, we move on to studying stability of critical points in the next part.

*Step 4. Unstable direction of the $\theta$ part*

Consider the dynamics in the form

$$\dot{\theta}_j = -\frac{1}{q_j(\tau) g_j(\Theta)} \nabla_{\theta_j} E(\Theta), \qquad \dot{q}_j = f_j(\Theta) - q_j,$$

where $g_j = Z_j$ and $f_j = \langle \theta_j, A_j(\Theta) \rangle$ for **Pre-LN**. In order to show that all limiting points $(\Theta^*, q^*), q^* > 0$ that are not fully synchronized (i.e. not all $\theta_j$ are equal) have measure zero basin of attraction, we only need to check that they are all unstable. More specifically, that the Jacobian matrix at any such point $\Theta^*, q^*$ has an eigenvalue with a positive real part. Because of the specific form of our system, the Jacobian has a convenient block form

$$J = \begin{pmatrix} J_{qq} & J_{q\theta} \\ J_{\theta q} & J_{\theta\theta} \end{pmatrix}$$

where $J_{qq} = -I_n$ and $J_{\theta q} = 0$ because at the critical point $\nabla_{\theta_j} E(\Theta^*) = 0$. Therefore, it is enough to show that $J_{\theta\theta}$ has a positive eigenvalue. Because of the gradient-like structure, $J_{\theta\theta}$ is the product of two matrices – $\mathrm{diag}(\frac{1}{f_1(\Theta^*)g_1(\Theta^*)}, \ldots, \frac{1}{f_n(\Theta^*)g_n(\Theta^*)})$ and a symmetric Hessian of the energy function $E$. The Hessian itself is unstable, this is an established result due to Criscitiello et al. (2024) (for $d \geq 3$) and Polyanskiy et al. (2025) (for $d = 2$) that together closed synchronization for **Post-LN**.

Surprisingly, this is enough for our cause, because of the following matrix property, that shows the product of the diagonal matrix and unstable Hessian is again unstable. Note that the lemma is not true without the symmetry assumption on $A$.

**Lemma 2.** *For a symmetric unstable matrix $A$ and a symmetric positive-definite $D$, the product $DA$ is also unstable.*

*Proof.* First, because $D$ is symmetric positive-definite, there is a symmetric positive-definite square root $P$, i.e. $P^2 = D$. Consider a symmetric matrix $B = PAP$. Notice that

$$P^{-1}DAP = PAP = B,$$

thus matrices $DA$ and $B$ are similar, i.e. they share eigenvalues. On the other hand, by Sylvester's law of inertia, $B$ and $A$ have the same inertia, and in particular the number of positive eigenvalues. Therefore, $B$ has a positive eigenvalue, and so does $DA$. □

*Step 5. Basin of attraction of unstable critical points.*

It is well-known that the set of unstable critical points of a dynamical system on a compact has measure-zero basin of attraction (see for example (Geshkovski et al., 2025, Lemma A.1) for a proof outline). Thus, we obtain that the event

$$\{\forall j \in [n] \; \forall t > 0 \; q_j > \frac{1}{m}\} \cup \{\forall j \in [n] \; \theta_j \to \theta_j^* \mid \Theta^* \text{is not synchronized}\}$$

has measure zero.

For completeness, we include the proof here. Let $\Phi_\tau(x_0)$ be the flow for the system $\dot{x} = F(x)$, where $x = (\Theta, q)$. The vector field $F(x)$ is smooth on the open domain where all $q_j > 0$. For any fixed $m > 0$, we consider the dynamics on the compact manifold

$$M_m := (S^{d-1})^n \times [1/m, Q]^n,$$

on which the flow is smooth. Let $K_m \subseteq M_m$ be the compact, forward-invariant set of initial conditions whose trajectories remain in $M_m$.

Let $S_{ns} \subset K_m$ be the set of non-synchronized critical points. By Step 4, every point $x^* \in S_{ns}$ is unstable. Let $A_{m,ns} \subset K_m$ be the basin of attraction for $S_{ns}$, i.e., the set of $x_0 \in K_m$ such that

$$\lim_{\tau \to \infty} \Phi_\tau(x_0) \in S_{ns}.$$

For any $x^* \in S_{ns}$, which lies in the interior of $M_m$, the Center-Stable Manifold Theorem applies. It guarantees the existence of a local center-stable manifold $W_{cs}^{\text{loc}}(x^*)$. The instability of $x^*$ implies that

$$\dim(W_{cs}^{\text{loc}}(x^*)) \leq \dim(M_m) - 1,$$

so $W_{cs}^{\text{loc}}(x^*)$ has measure zero. From the Center-Stable Manifold Theorem, there is a neighborhood of $x^*$ such that any trajectory staying in this neighborhood has to enter and remain on $W_{cs}^{\text{loc}}(x^*)$. By choosing a finite covering of the compact set $S_{ns}$ with respective neighborhoods of $x^*$, we get that any initial condition $x_0 \in A_{m,ns}$ has a trajectory $\Phi_\tau(x_0)$ that must eventually enter and remain on some $W_{cs}^{\text{loc}}(x_k^*)$, with a finite number of $x_k^*, k \leq K$ chosen from the covering. Thus, for some $N \in \mathbb{Z}_+, k \leq K$, we have

$$x_0 = \Phi_{-N}(\Phi_N(x_0)), \quad \text{where } \Phi_N(x_0) \in W_{cs}^{\text{loc}}(x_k^*).$$

Since $\Phi_{-N}$ is a local diffeomorphism, it preserves the dimensionality. Manifold $W_{cs}^{\text{loc}}(x_k^*)$ has positive co-dimension, and thus its pre-image too, which implies that it has measure zero in $M_m$. Consequently, measure of $A_{m,ns}$ is also zero, as a countable union of measure zero sets. Finally, to completely finish, we need to map the set to $t = 0$, because $\tau = 0$ corresponds to $t = 1$. This is again a smooth backward flow that preserves measure zero set. We arrive at measure zero set in initial coordinates $(\Theta(0), r(0))$, because they are distributed with standard Gaussian $r(0) \cdot \Theta(0)$.

□

**Remark 2.** *Here we describe modifications of the proof for each scheme.*

- ***Post-LN*** *No time change is required. The system is already autonomous and compact. Step 2 works with modified gradient descent from 1, because the modification matrix $M(t)$ is diagonal with blocks $\frac{1}{Z_j}$, that are uniformly bounded. As such, we get convergence to some critical point. Finally, we use existing analysis of the stability of the energy functional together with 2 to establish synchronization.*

- **Peri-LN** *For the Step 1 we use time change $\tau := \ln t$ and also consider $q_j(t) := r_j(t)/t$. It leads to the dynamics of the form*

$$\dot{\theta}_j = \frac{1}{q_j \|A_j(\Theta)\|} P_j A_j(\Theta), \qquad \dot{q}_j = \frac{\langle \theta_j, A_j(\Theta) \rangle}{\|A_j(\Theta)\|} - q_j.$$

  *The rest of the proof remains the same as **Pre-LN**, because this system satisfies gradient-like structure of Step 2, we also use the assumption to separate $q_j$ from 0, and then show that all critical points that are not synchronized have unstable direction in Step 4. The form of the system and the Jacobian in Step 4 is written generally, to accommodate this case too.*

- **Mix-LN** *At infinity **Mix-LN** follows exactly **Pre-LN**, and the argument follows from the proof of **Pre-LN**.*

- **nGPT** *Time change $\tau = \int \alpha_t^{-1}$ from Step 1 simplifies **nGPT** to the case $\alpha_t \equiv 1$. This makes the original dynamics autonomous on a compact manifold. As such, it requires no frame change, and we immediately move on to studying convergence and local behavior of that system. Modified Łojasiewicz from Step 2 and analysis of the unstable direction of the Jacobian from Step 4 follow similar steps. For the Jacobian, the only component is $J_{\theta\theta}$, and it is unstable from the same Lemma. The only complication for the system are points with $A_j = 0$. They, however, break the original dynamics too, and can be excluded with careful analysis.*

- **sqrt-scaling** *Time change from Step 1 with $\tau = 2\sqrt{t+1}$ reduces **sqrt-scaling** to **Post-LN**.*

## E  Simulation results with random weight matrices

**Attention Update Formulation.** To align our simulations with practical transformer architectures, we now explicitly include the output projection matrix, $W \in \mathbb{R}^{d \times d}$, in the attention update. For a multi-head configuration, the output of each head $h$ is first computed and then concatenated, after which the final projection is applied: $O_h = \text{softmax}(\beta X Q_h K_h^T X^T) X V_h, h = 1, \ldots, n_{\text{heads}}$ $X_{t+1} = \text{Concat}(O_1, \ldots, O_{n_{\text{heads}}})W$ where $Q_h, K_h, V_h \in \mathbb{R}^{d \times d_{\text{head}}}$. The inclusion of the matrix $W$ is a linear transformation applied after the core softmax-driven interaction. While this is crucial for model capacity in practice, it does not impact the theoretical dynamics description, which is why it was omitted from the preceding theoretical analysis for notational simplicity.

**Experimental Settings.** We present simulation results illustrating the evolution of average token cosine similarity. All plots show the mean trajectory averaged over $10^5$ independent runs, with shaded regions indicating the 90% confidence interval. Each run begins with a fresh draw of initial token positions $X$ from an isotropic distribution and random weight matrices. All simulations use a context of $n = 128$ tokens. For the normalization methods **Mix-LN** and **nGPT**, we use parameters $\tau = 0.25T$ and $\alpha \equiv 1$, respectively.

Our plots vary several factors. The majority of our experiments use Kaiming initialization. In this setting, we fix the number of heads to $n_{\text{heads}} = 1$ (so $d_{\text{head}} = d$) to isolate the core dynamics. We systematically vary the following parameters:

- **Dimension** ($d$): small (16), medium (128), and large (512).
- **Temperature** ($\beta$): low ($\beta = 1$), medium ($\beta = \sqrt{d}$), and high ($\beta = 4\sqrt{d}$).
- **Weight Sampling**: *static* (a single draw of $Q, K, V, W$ fixed for all time steps) vs. *resampled* (new matrices are drawn at each time step $\Delta t$).

**GPT-style Initialization**: We conduct one experiment that mirrors the configuration of a small GPT-2 style model.

- It uses $d = 768$, $n_{\text{heads}} = 12$ (implying $d_{\text{head}} = 64$), and a temperature of $\beta = \sqrt{d_{\text{head}}}$.
- Weights are drawn from a Gaussian distribution with variance $\sigma^2 = 0.02$ and are held *static*.

Figures are arranged to facilitate comparison, with each caption specifying the experimental signature $\langle d, n_{\text{heads}}, \beta, \text{weights}, \text{init} \rangle$.

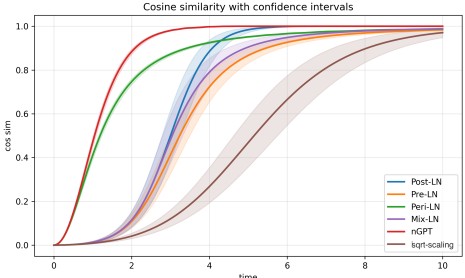

(a) $d = 512, n_{\text{heads}} = 1, \beta = \sqrt{d}$ (**medium**), static Kaiming weights. Case where $d > n$.

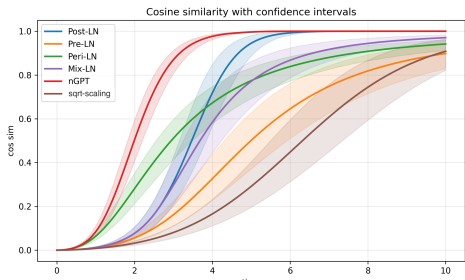

(b) $d = 512, n_{\text{heads}} = 1, \beta = 4\sqrt{d}$ (**high**), static Kaiming weights. Case where $d > n$.

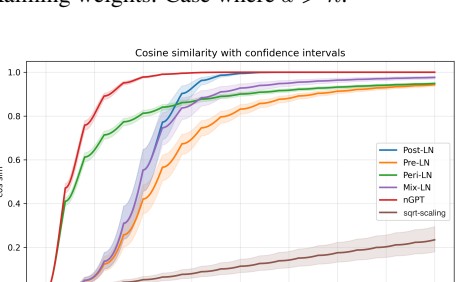

(c) $d = 512, n_{\text{heads}} = 1, \beta = \sqrt{d}$ (medium), **re-sampled** Kaiming weights at each $\Delta t = 1$.

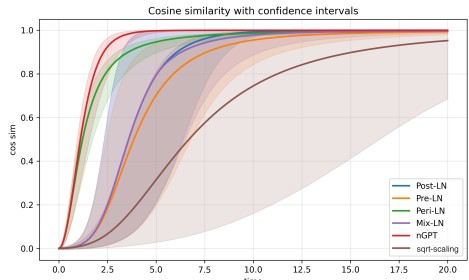

(d) $d = 16, n_{\text{heads}} = 1, \beta = 1$, static Kaiming weights. Case where $d < n$.

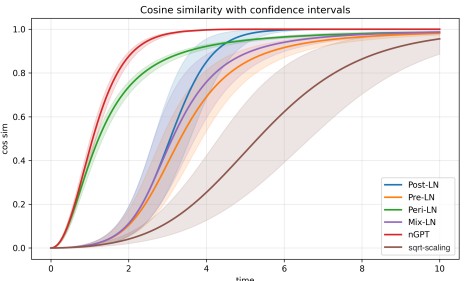

(e) $d = 128, n_{\text{heads}} = 1, \beta = \sqrt{d}$ (**medium**), static Kaiming weights. Case where $d = n$.

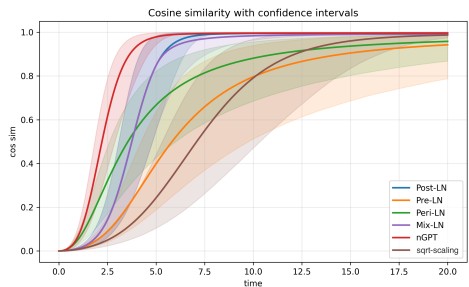

(f) $d = 128, n_{\text{heads}} = 1, \beta = 4\sqrt{d}$ (**high**), static Kaiming weights. Case where $d = n$.

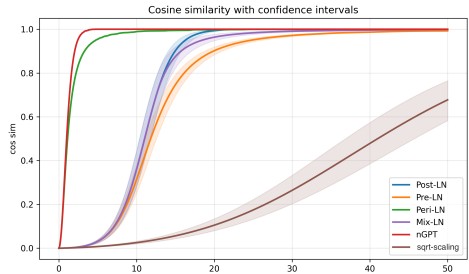

(g) **NanoGPT-style**: $d = 768, n_{\text{heads}} = 12$ ($d_{\text{head}} = 64$), $\beta = \sqrt{d_{\text{head}}}$, static Gaussian weights with $\sigma = 0.02$.

Figure 5: Evolution of average cosine similarity for tokens under the pure attention update.

