# OpenReview forum: "Normalization in Attention Dynamics"
_NeurIPS.cc/2025/Conference — NeurIPS 2025 poster_

### Official Review · Reviewer_bGjp · 2025-07-03

**Clarity:** 3
**Significance:** 3
**Originality:** 3
**Rating:** 4
**Confidence:** 1

**Summary:**

This paper presents a very interesting analysis on the normalziation layers in transformers. The authors model the evolution of tokens as interacting particles on a sphere and show that normalization acts as speed regularizaiton. With this, they analyze several normalization schemes and provide recommendations along different axes.

**Questions:**

1. While the paper analyses only on initial and terminal speeds, what can the authors comment about the intermediate region? The authors ackowledge this limitation. But, I would like to hear the thoughts that authiors have on how this can be analyzed.
2.

**Ethical Concerns:**

["NO or VERY MINOR ethics concerns only"]

**Final Justification:**

I am not a theory person, so I am not so confident about the theory results. But, the results look interesting, hence I stay with my initial rating.

**Limitations:**

I am not an expert in theory, so it is a bit hard for me to fully address the limitations of this work. Looking at a high level, the analysis looked interesting.

**Paper Formatting Concerns:**

-

**Quality:**

3

**Strengths And Weaknesses:**

Strength:
1. The theoretical study is quite interesting. I like the idea of analyzing the impact of normalization layers. The field of transformers is quite empirical and such analysis is quite useful to the community.
2. I like how the authors provide recommendations after each theorem.
- In 4.1, they suggest that Pre-LN, Mix-LN, and nGPT cluster more gradually than their counterparts, indicating a more effective use of intermediate layers.
- In 4.2, they show that Peri-LN/nGPT layers move tokens order-one distances on the hypersphere, while Post-LN and Pre-LN advance more slowly.
- In 4.3 they say that  Post-LN and LN-Scaling cluster toke directions at an exponential rate, while Pre-LN, Peri-LN and Mix-LN slow down to a polynomial decay.
With these they provide recommendation that  Peri-LN and  nGPT seems to be the better choice. Such kind of recommendation is quite interesting.

Weakness:
1. It would have been nice if the theoretical explanations and observations were backed by some expeirments. If the authors showed that some LLM training on small dataset indeed produces these observations, that would have backed their claims even strongly.
2. The assumption that the Q, K and V matrices are identity is quite strong.

---

> ### Author Rebuttal · Authors · 2025-07-31
>
> We thank the reviewer for recognizing our theoretical contributions and for appreciating the practical recommendations we derive from our analysis. We're glad you found the insights about Pre-LN, Peri-LN, and nGPT's relative advantages well-structured.
>
> **On experimental validation:**
> We understand the desire for empirical backing. We'd like to clarify that the normalization schemes we analyze (Post-LN, Pre-LN, Peri-LN, etc.) have already been extensively validated empirically:
> - **Pre-LN vs Post-LN**: Xiong et al. (2020) demonstrated Pre-LN's superior training stability
> - **Peri-LN**: Kim et al. (2025) showed its effectiveness. This approach was recently adopted by Gemma 3 and a slightly modified version by OLMo 2. As we establish its effectiveness theoretically, it is nice to see how relevant it becomes in practice.
> - **nGPT**: Loshchilov et al. (2025) validated its performance with learnable α_t
>
> Our contribution is providing theoretical understanding of *why* these empirically successful methods work. Rather than proposing untested schemes, we explain the mechanisms behind proven approaches. It would be extremely interesting to experiment with custom normalizators based on our study, but it is a complex practical contribution to devise a novel normalization approach. As such, we think it is out of scope of this primarily theoretical paper, but would be a great avenue for future research.
>
> **On the Q=K=V=I assumption:**
> We appreciate this concern and want to clarify:
> - **Theorem 4.2** actually handles **arbitrary $Q,K,V$** (only requiring $||Q^TK||, ||V|| ≤ 1$)
> - **Theorem 4.3** uses arbitrary $Q,K$ matrices, and $V = I$. Moreover, this is done to focus on the clustering introduced by attention, whereas adding $V$ would add some drift into the dynamics. In other words, tokens would simultaneously cluster and move, and the intuition derived from this result remains, but it would not be as clean.
> - **Theorem 3.1** uses $Q = K = V=I$ only for convenience of exposition (because the proof is quite complicated), but the proof works for a more general $Q^\top K = Diag(\lambda_i) V, \lambda_i > 0$ case. This generalises even prior works that study pure postln, because they assume $Q^\top K = V$ (see Geshkovski et. al.)
> - Only Theorem 4.1 uses $Q=K=V=I$, but it serves as an analytical baseline to establish what to expect.
>
> **On limitations**
> Additionally, we have updated Figure 4 to shows that qualitatevely the dynamics of Figure 2a, modeled by Theorem 4.1 is the same in much more general case: with Kaiming-initialized random matrices, multiple attention heads, and time-varying parameters. This, effectively, allows for exact parameters employed in real transformers. This backs up our claim that introduced assumptions on $Q, K, V$ are not that restrictive.
>
> Regarding MLPs: These add token-wise transformations that don't influence the mixing component, each token additionally has its own drift. This effect does not break our relative comparisons between schemes, but it complicates overall study. We leave it for future research.
>
> **On intermediate region analysis (Question 1):**
> Great question! Analyzing the intermediate regime is the most complicated topic in this field. Relative to our work, it is not clear how speed regulation affects the space component where tokens end up. And starting from different parts of the state space the dynamics can be different. However, there are prominent works (for example Bruno et. al.) that study initial and intermediate regimes, most notably focusing to dimension reduction in the initial regime and meta-clustering in the intermediate regime. In their analysis they use PostLN. We think that our framework can be combined with those approaches, to derive similar results(with say different rates of convergence). This is a great topic for future research.
>
> Thank you for your supportive review and for highlighting areas where our exposition could be clearer.

---

### Official Review · Reviewer_rFjq · 2025-07-03

**Clarity:** 3
**Significance:** 2
**Originality:** 2
**Rating:** 2
**Confidence:** 3

**Summary:**

This paper provides a theoretical analysis of various normalization schemes in transformer architectures by modeling token representations as interacting particles on a sphere. The authors reframe different normalization methods (Post-LN, Pre-LN, Mix-LN, Peri-LN, nGPT, and LN-Scaling) as variations of a unified interacting particle system where normalization acts as speed regulation. Through this framework, they analyze how different schemes affect clustering dynamics and representation collapse, establishing asymptotic clustering results and characterizing initial and terminal token velocities. The work identifies Peri-LN as particularly effective for maintaining meaningful token evolution across both early and deep layers.

**Questions:**

1. How do the theoretical predictions about different normalization schemes translate to measurable differences in LLM capabilities? Could you provide experiments showing how Post-LN vs Pre-LN vs Peri-LN affect reasoning ability, language understanding, or other downstream tasks? This connection between your mathematical framework and real model performance would significantly strengthen the practical relevance of your work.
2. Given the substantial assumptions (Q = K = V = I_d, no MLP layers), what is your roadmap for extending this analysis to realistic transformer architectures? Can you provide preliminary results or concrete next steps that would make your theoretical framework applicable to actual LLMs used in practice?

**Ethical Concerns:**

["NO or VERY MINOR ethics concerns only"]

**Limitations:**

Yes

**Quality:**

2

**Strengths And Weaknesses:**

Strengths
1. The paper provides an elegant unification of diverse normalization schemes under a single dynamical systems perspective. By casting all methods as speed regulation mechanisms for particles on a sphere, the authors enable direct theoretical comparison of fundamentally different approaches.
2. The theoretical results are well-developed, particularly Theorems 3.1-4.3, which establish asymptotic clustering guarantees and characterize velocity profiles. The extension of gradient flow analysis to general normalization schemes beyond Post-LN is a significant theoretical contribution.
Weaknesses
1. The analysis relies on highly restrictive assumptions (Q = K = V = I_d, omission of MLP layers, simplified attention) that significantly limit its applicability to real transformer architectures. The gap between theory and practice is substantial, making it difficult to extract actionable insights for actual model design.
2. The paper fails to demonstrate how their theoretical insights about clustering dynamics and representation collapse translate to actual LLM performance on reasoning tasks, language understanding, or other practical capabilities. Without showing the impact on real-world model behavior, the theoretical contributions remain disconnected from what practitioners care about most.

---

> ### Author Rebuttal · Authors · 2025-07-31
>
> We thank the reviewer for recognizing our "elegant unification of normalization schemes" and "well-developed theoretical results." We're happy to clarify how our work bridges theory and practice.
>
> **On assumptions - less restrictive than they appear:**
> Our theoretical framework follows established conventions (Geshkovski et al., Sander et al.) while actually being quite general:
>
> - **Theorem 4.2**: Handles **arbitrary $Q,K,V$** (only requiring $||Q^TK||, ||V|| ≤ 1$)
> - **Theorem 4.3**: Has **arbitrary $Q, K$** with the only assumption $V = I_d$, to remove the drift and focus on clustering
> - **Theorem 3.1**: Can be extended to **$Q^TK = Diag(\lambda_i)V$**, generalizing all prior work
>
> We assume $Q=K=V=I_d$ in Theorem 3.1 for simplicity of exposition, so that the presented proof is not too overwhelming.
>
> The **Theorem 4.1** with $Q=K=V=I_d$ case serves as an analytical baseline. We have updated Figure 4 to further show that our predictions (modeled in Figure 2a) can be observed with:
> - Kaiming-initialized random matrices
> - Multiple attention heads
> - Time-varying matrices
> - Various $\beta$
>
> Regarding MLPs: These layers add token-wise transformations that don't affect relative comparisons between schemes that significantly. Including them would obscure the key insights about how normalization specifically impacts token clustering, guided by attention mechanism. This simplification is a deliberate choice following previous established theoretical analysis (Geshkovski et al., 2023, 2025).
>
> Answering **Question 1**, the assumptions on matrices $(Q, K, V)$ are already not that restrictive, with the figures supporting that even more general changes (varying common architecture parameters) do not change the presented qualitative results. The introduction of MLP layers is an interesting avenue for further work (currently there are no proved clustering results with MLP layers), but it is not that significant for the work presented. MLP layers add a drift term to the dynamics, and one can still compare different normalization schemes with that term, but the impact on the **clustering** would be not as easy to see.
>
> **On practical relevance and empirical experiments:**
>
> The reviewer asked about experiments comparing normalization schemes. We build upon extensive existing empirical work -- each studied scheme was introduced/analyzed with an empirical paper justifying its performance, for example:
> - Xiong et al. (2020): Pre-LN outperforms Post-LN
> - Kim et al. (2025): Peri-LN's effectiveness, the scheme adopted in Gemma 3
>
> **Our unique contribution for practitioners**: We explain *why* some of these empirical phenomena occur:
> - Post-LN's exponential clustering → representation collapse
> - Pre-LN's polynomial clustering → better depth utilization
> - Peri-LN's balances initial and late velocities → optimal performance
>
> This isn't just retrospective analysis, as originally our work was motivated by bridging recent Self-Attention dynamics research with a widely adopted practical usage of PreLN. However, the analysis presented showed clear advantages of PeriLN architecture. It was later we learned about its adoption in production models like Gemma-3 and (with modifications) OLMo-2.
>
> **On bridging theory and practice:**
>
> Our work does not describe full Transformer in all details, but it does not make it completely "disconnected from practitioners". We aim to connect the evolving theoretical field of transformers (started with works by Geshkovski et.al.) with the components important in practice. However, already the simplified analysis introduced, focusing on a specific architecture component, allowed to explain phenomena practitioners observe (representation collapse), and predict efficiency of PreLN compared to PostLN, and PeriLN in general, in a one unified approach.
>
> **On theoretical contributions in ML:**
>
> We respectfully disagree that theoretical work requires new empirical validation to be valuable. The transformer's core architecture (attention, residuals, normalization) survived the test of time in a rapidly progressing practical field. Understanding *why* it works is important in itself, and it could enable principled design improvements rather than trial-and-error in any future research.
>
> However, experimenting with **novel** design choices based on our analysis asks for pretraining experiments, which is hard and computationally requiring. This makes it a topic for a separate paper. Existing methods already have supporting experiments, and our study shows another argument to choose between them.
>
> **Looking forward:**
>
> This work opens exciting avenues:
> - Designing custom normalization
> - Choosing appropriate normalization for different model depths
> - Extending existing theoretical analysis to include the impact of normalization schemes.
>
> We hope this clarifies that our work aims to create a clear and unified theoretical perspective on the impact of normalization layers onto geometry of tokens, helping both bridge an evolving theoretical field with practice and empower more informed decisions for practicioners.

---

> > ### Author Response · Authors · 2025-08-06
> >
> > Dear Reviewers, given that the rebuttal period is going to close pretty soon, we would like to ask whether our responses fully addressed your concerns. Please do not hesitate to send us more questions or requests for clarification.

---

### Official Review · Reviewer_KEPZ · 2025-07-03

**Clarity:** 3
**Significance:** 3
**Originality:** 4
**Rating:** 5
**Confidence:** 3

**Summary:**

This paper presents a theoretical analysis of different layer normalization schemes in transformers by modeling the evolution of token representations across layers as a continuous-time interacting particle system. Within this framework, various normalization methods are characterized by a “speed regulation” factor that modulates the dynamics and affects the phenomenon of representation collapse. By comparing the speeds at early and later layers, the authors identify Peri-LN as a favorable compromise.

**Questions:**

- Cited works (e.g., Geshkovski et al., 2025, Figure 2) exclude the d=2 and intermediate beta case from their clustering theorems. Does Theorem 3.1 guarantee clustering even in these regimes? Furthermore, to improve clarity for the reader, it would be beneficial to add a reference in the main text to the discussion in the appendix regarding the assumption s_j' ≥ c > 0, as it appears to be a key condition for the theorem.
- Generalizing the results to arbitrary $Q, K, V$ matrices is clearly challenging. However, considering the simple case where $Q=K=I_d$ and $V=-I_d$ could provide valuable insight. In this setting, the dynamics become repulsive, encouraging tokens to spread apart rather than cluster. Do the paper's conclusions about the effectiveness of different normalization schemes, particularly the advantages of Peri-LN, still hold in this repulsive regime?
- The inclusion of an MLP layer would introduce a drift term into the dynamics, independent of the attention mechanism, which would likely have a strong effect on $r_j(t)$. What is the authors' intuition on how this MLP-induced drift would interact with the different normalization schemes?

**Ethical Concerns:**

["NO or VERY MINOR ethics concerns only"]

**Final Justification:**

Assuming that the additional experiments discussed by the authors in the rebuttal are consistent with their claims, and that the final version will strengthen the connection to relevant results in the literature, particularly in the supplementary material, I believe this paper could help increase the community’s interest in an emerging mathematical framework for studying transformers, while also demonstrating a concrete application. This is despite the existing limitations, which I recommended the authors clearly state in the paper. For these reasons, my final score is an accept.

**Limitations:**

Yes

**Quality:**

3

**Strengths And Weaknesses:**

**Strengths:**
- The paper's main strength is its unification of various normalization schemes into a single dynamical systems framework. By modeling different schemes as "speed regulations", it provides an intuitive basis for comparison.

- This framework explains several empirically observed phenomena, such as the superior stability of Pre-LN over Post-LN in deep networks. The analysis of initial and terminal velocities offers insights into how different schemes utilize early versus late layers, leading to actionable conclusions about architectural design.

- The work rigorously extends a recent line of theoretical analysis (Geshkovski et al.). While based on a simplified model, the paper's conclusions are theoretically sound and provide an interesting starting point for future research to build upon, potentially by relaxing assumptions or incorporating more architectural components.

**Weaknesses:**
- The model is still quite simplified (see questions 1 and 2 below). I understand that it is mathematically very challenging to extend the results, but some more discussion about different values for Q, K, V and the inclusion of the MLP could help the reader.
- (Minor) The presentation is clear but there are several unclear phrases or typos, for example:
  - Line 91: At(X) = [X11(X), . . . , Atn(X)]
  - Line 122: "we readily get: Note that"
  - Line 126: "(a.k.a)"
  - Line 136: "I particular"
  - Line 214: "which tokens a pre-clustered"

---

> ### Author Rebuttal · Authors · 2025-07-31
>
> We sincerely thank the reviewer for their careful reading and insightful questions. Your technical understanding of our framework is much appreciated.
>
> **On typos and presentation:**
> Thank you for identifying these errors. We will fix all of them in the revision.
>
> **On clustering in $d=2$ and intermediate $\beta$ (Question 1):**
> Great question! Yes, Theorem 3.1 works in these regimes, because it uses a very recent result, that completed the proof for $d=2$. We clearly state that in the revised appendix.
>
> Thank you for the suggestion, we are going to add a discussion about that condition. It is mainly technical, and we will present it in the way easiest to analyze. Actually, the condition can be omited for PostLN, nGPT, LN-Scaling, and modified to $\dot r_j ≥ c > 0$ for PreLN and PeriLN, which is easier to understand given the Table 2 definition, we'll modify it for clarity.
>
> **On assumptions - less restrictive than they appear:**
>
> Our theoretical framework follows established conventions (Geshkovski et al., Sander et al.), but in that scope it is actually quite general:
>
> - **Theorem 4.2**: Handles **arbitrary $Q,K,V$** (only requiring $||Q^TK||, ||V|| ≤ 1$)
> - **Theorem 4.3**: Arbitrary attention with $V = I_d$
> - **Theorem 3.1**: Can be immediately extended to $Q^\top K = Diag(\lambda_i)V, \lambda>0$, generalizing even prior work (Geshkovski et. al. assume $Q^\top K = V$). The case $Q=K=V=I_d$ is presented to simplify exposition of an already complicated proof.
>
> The **Theorem 4.1** with Q=K=V=Id case serves as an analytical baseline, later modeled in Figure 2a. We updated Figure 4 to model the dynamics with:
> - Kaiming-initialized random matrices
> - Multiple attention heads
> - Time-varying matrices
> - Various temperature parameters β
>
> In all those cases the qualitative behaviour is the same as in Figure 2a (current Figure 4 can be misleading, because it uses $d=2$).
>
> Regarding MLPs: These add token-wise transformations that don't affect relative comparisons between schemes too much. Including them would be an interesting line of research, but it also might obscure the key insights about how normalization specifically impacts token mixing. We think this makes it a topic for another work.
>
>
> **On repulsive dynamics with V=-I (Question 2):**
> Fascinating question! With V=-I, the dynamics is governed with repulsive interactions between tokens, instead of attracting ones. Let us quickly note, that empirically the heads with negative $V$ matrices are rare (though exist), arguably their behaviour is to mitigate representation collapse, and in this scope they should be studied as a part of the system. Analyzing pure $V = -I_d$ case is possible in our framework, but it does not directly fit the clustering perspective (Theorem 3.1 and Theorem 4.3), and changes the system in more ways than one, because token magnitudes $r_j$ would evolve differently. As such, it is not clear if in this case PeriLN remains the best choice, but practically it could appear as only one head out of many, not having that drastic impact (thus PeriLN still would be best).
>
> **On MLP-induced drift (Question 3):**
> As correctly noted, MLPs add a drift term to the dynamics. Indeed, it has a different impact on the dynamics, introducing additional attraction in specific directions. This term does not impact the evolution of magnitudes $r_j$ drastically, the order of growth remains the same. As such, the core intuition behind comparing all schemes persists.
>
> **Final words**
> All in all, we want to highlight that our assumptions might appear more restrictive than they are, especially when compared to common practice in the field. We use $Q=K=V=I_d$ in Theorem 3.1 for convenience, while the proof is correct in a more general case. The same assumption is made in Theorem 4.1 because it serves as a complete analytical baseline, while Theorems 4.2-4.3 are much more general.
>
> Other than that, thank you for all the questions and highlighted points for improvement, we hope we answered the questions and we are going to use all suggestions in the revision.

---

> > ### Comment · Reviewer_KEPZ · 2025-08-05
> >
> > I would like to thank the authors for their responses during the rebuttal and for the additional numerical experiments. Regarding Theorem 3.1, I recommend that the proof refer more precisely to the external results it builds upon, as the current references are somewhat vague, particularly if the authors intend to include the remaining cases for $Q$, $K$, and $V$.
> >
> > Despite the current limitations, I find the authors’ mathematical approach to studying normalization both original and insightful. Assuming the additional experiments are representative and that the final version will more transparently acknowledge the limitations of the results, I am increasing my score to 5.

---

### Official Review · Reviewer_cqGC · 2025-07-05

**Clarity:** 3
**Significance:** 3
**Originality:** 3
**Rating:** 6
**Confidence:** 1

**Summary:**

They conduct a theoretical analysis of how the radial component of token representations evolves with transformer depth and as a function of the layer normalization scheme used. They compare several different layer norms: pre-LN, post-LN, Peri-LN, nGPT, Mix-LN, and LN-Scaling. They find that Pre-LN makes better use of depth, as tokens continue to evolve across many layers. Moreover, nGPT has the advantage of being able to adapt the rate of representation evolution / clustering.

**Questions:**

Is it reasonable for future work to conduct similar analysis without as many simplifying assumptions on the architecture e.g. no MLP / restricting the attention?

It seems that a lot of your findings match empirical beliefs in the field, are there any results where there is a mismatch from empirically derived preferences practitioners have for LN schemes?

**Ethical Concerns:**

["NO or VERY MINOR ethics concerns only"]

**Final Justification:**

I appreciate the author's response and it seems like solid work, but my ability to confidently asses the work is low as it is outside my main area of research.

**Limitations:**

One thing that seems to be missing is a prediction or improvement to existing methods that would result from this analysis. Otherwise the authors seem to do a good job of listing out limitations in the paper.

**Paper Formatting Concerns:**

It looks good!

**Quality:**

3

**Strengths And Weaknesses:**

This paper is a bit outside the area of research I have worked in for the last several years, so this one is pretty difficult for me to evaluate properly. I don't have the ability to rigorously evaluate the theory, which is probably the most important piece here. And as for the validity of their simplifying assumptions, I don't have a strong sense here either (e.g. removing MLP and restricting the attention KQV).

I will say that the paper seems to be well written and directionally I think it is important to have a stronger understanding of how these decisions shape the NN dynamics.

The takeaways from the paper seem to match empirical findings (e.g. Pre-LN is preferable to Post-LN), which is a good sign.

One thing that seems to be missing is potentially a prediction or improvement to existing methods that would result from this analysis.

---

> ### Author Rebuttal · Authors · 2025-07-31
>
> We sincerely thank the reviewer for the strong support and for recognizing the importance of understanding how normalization decisions shape neural network dynamics.
>
> **On assumptions:**
> Our assumptions follow field conventions (Geshkovski et al.) but are actually quite flexible:
> - **Theorem 4.2**: Handles **arbitrary Q,K,V**
> - **Theorem 4.3**: Arbitrary Q,K with V=I_d
> - **Theorem 3.1**: Presented with Q=K=V=I_d for clarity, but extends to Q^TK=Diag(λ_i)V
>
> In the revision we updated Figure 4 to validate predictions with Kaiming-initialized random matrices, multiple heads, and varying parameters - showing our insights generalize beyond the theoretical assumptions.
>
> Regarding MLPs: Following standard practice, we focus on attention-driven dynamics. MLPs add drift terms that don't fundamentally change normalization comparisons.
>
> **On predictions and improvements:**
> Our analysis yields concrete insights:
> 1. **Peri-LN is optimal** - balancing early/late layer dynamics (used by recent Gemma 3 and (with modification) OLMo 2)
> 2. **Design principle** - Future architectures could use custom speed regulation s_j(t), or employ the framework to help with the design. For example, current nGPT paper uses trainable $\alpha_t$, with initial value empirically picked. However, their initial value would work poorly at scale, whereas from introduced analysis they could tailor what $\alpha_t$ to use at scale.
>
> **On empirical alignment:**
> Our findings confirm Pre-LN > Post-LN (well-known empiricaly wisdom) but also suggest Peri-LN > Pre-LN - a prediction validated by several recent production models, but still not widely adopted.
>
> **Looking forward:**
> We hope our framework would help to bridge the gap between the theory of token propagation in transformers and practically employed models. As noted, introducing the term induced by MLP layers is a natural step for further research(understanding its exact impact on the clustering behaviour).
>
> Thank you for your support and thoughtful questions!

---

### Note · Authors · 2025-08-14

Dear AC and reviewers,
We appreciate your feedback throughout the review process. Here's a brief overview of the rebuttal:
Our paper initially received scores of 6/4/4/2. Reviewers appreciated the novel theoretical framework but raised concerns about strict assumptions and practical relevance.

Reviewer KEPZ (4→5) raised valid questions we clarified in the rebuttal. Based on our discussion they increased the score.

Reviewers cqGC and bGjp did not respond, and we hope it means they decided to keep their positive marks (6 and 4).

Reviewer rFjq (2) did not answer to our rebuttal, even though they are the only one to give a negative score. Unfortunately, this gave us no room for clarification and addressing raised concerns in a discussion.

To underline our answers to most asked questions:
1. Our assumptions are less restrictive than initially perceived by reviewers (Q=K=V=I_d) — Theorems 4.2-4.3 handle much more general cases of Q,K,V, and proof of Theorem 3.1 relies only on standard assumptions for the field.
2. The work introduces an important practical component (Normalisation layer) into an evolving theoretical field of ode-based analysis of token geometry in transformers. Used assumptions (omission of MLPs, restriction on Q, K, V) are very common in that field.
3. We focus on a theoretical side of the topic, unifying and explaining why existing methods (Pre-LN, Peri-LN) work, and how to compare their influence on token geometry, not developing new practical methods. The experiments for the methods we analyze were already conducted in the literature.
4. Our analysis fits well-known(dominance of PreLN) and predicts just recently applied(efficiency of PeriLN) practical knowledge.

We hope that mostly positive feedback, together with increased score by KEPZ's after rebuttal, shows that our work makes valuable theoretical contributions within its intended scope. Thank you for your consideration.

---

### Decision · Program_Chairs · 2025-09-17

**Decision:**

Accept (poster)

**Comment:**

This paper studies
how different normalization schemes influence clustering dynamics and token representations
within a unified framework of interacting particle dynamics on the sphere and speed regulation formulation.


The reviewers appreciate the **strengths** of the paper:
(cqGC) well-written, takeaways seem to match empirical findings,
(KEPZ) unification of various normalization schemes into a single dynamical system framework, intuitive basis (speed regulations) for comparison,
Explains several empirically observed phenomena, rigorously extends a recent line of theoretical analysis,
(rFjq) elegant unification of diverse normalization schemes under a single dynamical systems perspective, theoretical comparison with speed regulation mechanisms, well-developed, significant theoretical contributions,
(bGjp) useful to the community, recommendations after each theorem.


The reviewers also find the **weaknesses** of the paper:
(cqGC) missing a prediction or improvement to existing methods,
(KEPZ) still quite simplified model, several unclear phrases,
(rFjq) highly restrictive assumption, fails to demonstrate how their theoretical insights translate to practical capabilities,
(bGjp) lack of experiments to support the theory, strong assumptions.

**After the discussion**, many reviewers were not very confident in their score, but they express a similar opinion: elegant unification and intuitive basis, but highly restrictive assumptions.
I recommend to accept this paper as it has significant merits despite the existing limitations (e.g., strong assumptions).